# In Silico Prediction of Steroids and Triterpenoids as Potential Regulators of Lipid Metabolism

**DOI:** 10.3390/md19110650

**Published:** 2021-11-22

**Authors:** Valery M. Dembitsky

**Affiliations:** Centre for Applied Research, Innovation and Entrepreneurship, Lethbridge College, 3000 College Drive South, Lethbridge, AB T1K 1L6, Canada; valery.dembitsky@lethbridgecollege.ca; Fax: +1-888-858-8517

**Keywords:** steroids, triterpenoids, hormones, regulators, lipid metabolism, activity

## Abstract

This review focuses on a rare group of steroids and triterpenoids that share common properties as regulators of lipid metabolism. This group of compounds is divided by the type of chemical structure, and they represent: aromatic steroids, steroid phosphate esters, highly oxygenated steroids such as steroid endoperoxides and hydroperoxides, α,β-epoxy steroids, and secosteroids. In addition, subgroups of carbon-bridged steroids, neo steroids, miscellaneous steroids, as well as synthetic steroids containing heteroatoms S (*epithio steroids*), Se (*selena steroids*), Te (*tellura steroids*), and At (*astatosteroids*) were presented. Natural steroids and triterpenoids have been found and identified from various sources such as marine sponges, soft corals, starfish, and other marine invertebrates. In addition, this group of rare lipids is found in fungi, fungal endophytes, and plants. The pharmacological profile of the presented steroids and triterpenoids was determined using the well-known computer program PASS, which is currently available online for all interested scientists and pharmacologists and is currently used by research teams from more than 130 countries of the world. Our attention has been focused on the biological activities of steroids and triterpenoids associated with the regulation of cholesterol metabolism and related processes such as anti-hyperlipoproteinemic activity, as well as the treatment of atherosclerosis, lipoprotein disorders, or inhibitors of cholesterol synthesis. In addition, individual steroids and triterpenoids were identified that demonstrated rare or unique biological activities such as treating neurodegenerative diseases, Alzheimer’s, and Parkinson’s diseases with a high degree of certainty over 95 percent. For individual steroids or triterpenoids or a group of compounds, 3D drawings of their predicted biological activities are presented.

## 1. Introduction

Lipids are a complex name for many metabolites of natural origin, and they are combined into various groups according to a common physical property, hydrophobicity, that is, insolubility in water [1,2,3,4]. However, this definition is currently not entirely correct, since some groups, phospholipids or sphingolipids and some others, manifest themselves as amphiphilic compounds; that is, they can dissolve both in polar substances and in non-polar solvents [5,6,7,8].

Lipids play an important role in the body in storing energy and are components of biological membranes, steroid hormones, bile acids, and vitamins. They come from food or de novo synthesis in the liver. Fatty acids, stored primarily as triglycerides, are the main source of energy for the muscles and heart. However, the overproduction and accumulation of triglycerides in adipose tissue and other tissues are closely associated with metabolic disorders in humans. Disorders of lipid metabolism lead to the development of many diseases, including atherosclerosis, which occurs because of a violation of cholesterol homeostasis and is closely associated with atherosclerosis [9,10]. In addition, estrogen and estrogen receptors are well-known regulators of several aspects of metabolism, including glucose and lipid metabolism, and impairment of estrogen signaling is associated with the development of metabolic diseases [11].

The regulation of lipid metabolism is of interest primarily in the context of the regulation of energy flow and the way of its integration with other energy sources in tissues. A special role in the regulation of lipid metabolism is played by hormones such as adrenaline and norepinephrine, glucagon, glucocorticoids, hormones of the anterior pituitary gland, as well as thyroxine and sex hormones [12,13,14,15,16,17,18,19,20].

This review is devoted to natural, semi-synthetic and synthetic steroids, and triterpenoids isolated from plants, fungi, marine invertebrates, and synthesized in various laboratories around the world. The presented class of molecules differs from many other steroids or triterpenoids that exhibit a wide range of biological activities aimed at lowering cholesterol by inhibiting cholesterol synthesis or other activities associated with these processes.

## 2. Aromatic Steroids Derived from Natural Sources

The most famous natural monoaromatic steroids that are of some practical interest in terms of regulators of lipid metabolism are estrone (**1**), estradiol (**2**), estriol (**3**), and equilin (**4**). The chemical structures of these aromatic steroids are shown in Figure 1. It is known that estrone is a minor female sex hormone, which was discovered in the 1920s from the urine of pregnant women independently by two groups of scientists from Germany and the United States, biochemist Adolph Butenandt and American scientists Edward Doisy and Edgar Allen, respectively [21,22,23,24,25,26,27,28]. Later, Adolf Frederik Johann Butenandt from the Institute of Chemistry in Götting (Germany) received the Nobel Prize in Chemistry in 1939 for the discovery of this hormone.

Estrone, or (8R,9S,13S,14S)-3-hydroxy-13-methyl-7,8,9,11,12,14,15,16-octahydro-6H-cyclopenta[a]phenanthren-17-one (**1**), is produced in vivo from androstenedione and/or testosterone via estradiol. The above-mentioned estrogens show estrogenic activity with various variations [29,30], although changes in the structure of estrogens D ring may also demonstrate anticancer activity [31]. The presence of female sex hormonal (**1**–**4**) estrogens was first detected in plants in 1926 by Dohrn and co-workers [32] and then, in the 1930s, simultaneously by Butenandt and Jacobi [26] and Skarzynski [33]. Recently, Janeczko and Skoczowski [34] published a survey in which they summarized the data on the presence of mammalian sex hormones and their physiological role in plants. These hormones, such as 3,17β-dihydroxy-1,3,5(10)-estratriene, rosterone, testosterone, or progesterone, were present in 60–80% of the plant species investigated. Butenandt and Jacobi [26] isolated estrone (**1**) from seeds and pollen of *Hyphaene thebaica*, *Glossostemon bruguieri*. *Glycyrrhiza glabra*, *Malus pumila*, *Phoenix dactylifera*, *Ph. vulgaris*, *Punica granatum*, *Salix caprea*, and *Salix* sp. The chemical structures of steroids and triterpenoids are shown in Figure 1, and biological activity is shown in Table 1.

From the 1930s to the present, more than 15,000 articles have been devoted to various issues of estrone and its derivatives, and more than 200 reviews summarize the data on the activity of this hormone. Estrone (**1**) and estradiol (**2**) and their derivatives (**3**–**10**) are the main natural estrogens found in humans and are involved in estrogen metabolism [35,36,37,38]. Endogenous estrogens in humans demonstrate a wide range of biological activities and are involved in the regulation of lipid metabolism, which has been the subject of tens of thousands of articles, reviews, and books. Additionally, in this short paragraph, we will not list and discuss this topic, but in Table 1, we give a short list of the main biological activities that aromatic steroids show.

Aromatic steroids found in animals and humans are also produced by some marine invertebrates and are found in the extracts of certain plants and fungi, as well as in marine sediments. Thus, the extract of bark from the main wooden rod of ketapang *Terminalia catappa* (family Combretaceae) contained estrone (**1**), estriol (**3**), equilin (**4**), equilin sulfate (**5**), and (**11**) [39]. The ethanol extract of leaves of *T. catappa* shows antimicrobial activity against *Escherichia coli*, *Staphylococcus aureus*, and *Candida albicans*, as well as antifungal activity against *Epidermophyton floccosum* and *C. albicans* [40,41,42]. In addition, extracts of leaves and roots and other parts have been studied for many medicinal properties and have shown antidiabetic, wound healing, anti-inflammatory, anticancer, antimicrobial, hepatoprotective, and antioxidant activities [43]. Unusual withanolide (20S,22R)-1,6β,17α,27-tetrahydroxy19-norwitha-1,3,5(10),24-tetraenolide, named jaborosalactone 7-(**12**), was isolated from the leaf extracts of *Juborosa leucotricha* [44].

The steroid 24,26-cyclo-19-norcholesta-1,3,5(10),22-tetraen-3-ol (**13**) is characterized by the presence of an aromatic A ring containing a cyclopropane ring in the side chain was isolated from the Hainan soft coral *Dendronephthya studeri* [45]. The ring A-aromatic bile acid 3-hydroxy-19-norchola-1,3,5(10),22-tetraen-24-oic acid (**14**) was isolated from methanol extract of the marine sponge *Sollasella moretonensis*, collected from the seabed of northern Queensland, Australia [46]. Commenting on the release of this unusual lipid, the authors point out that aromatic steroids are more typical for plants and less so for marine invertebrates or higher animals. A 4-hydroxy-6-oxopregnane-3-glycoside with an aromatic ring A containing the sugar 6′-deoxy-L-β-altropyranose-4′-acetate named hapaioside (**15**) was found in the extract of a Pohnpei, a tube sponge, *Cribrochalina olemda* from the family Niphatidae [47]. The activity of this glycoside has not been determined, although it is known that this marine sponge contains cyclic peptides called kapakahines, some of which show cytotoxicity against P388 murine leukemia cells [48]. Structurally similar to hapaioside, 3-(4-O-acetyl-6-deoxy-β-galactopyranosyloxy)-19-norpregna-1,3,5(10),20-tetraene (**16**) was isolated from bush coral *Alcyonium gracillimum* (syn. *Scleronephthya gracillimum*), which was collected from the Gulf of Sagami, Japan [49].

Monoaromatic B ring steroids are rare natural lipids found predominantly in the mushroom kingdom and are also present in marine sediments or oils. Thus, phycomysterol A (**17**) and C (**18**), which possess a unusual natural 19-norergostane skeleton with an aromatic B ring, are synthesized by a filamentous fungus *Phycomyces blakesleeanus* (order Mucorales, the phylum Zygomycota) [50]. In addition, phycomysterol A showed antiviral inhibition at a concentration of 0.64 µg per well (200 µL) and showed (IC_50_ = 5.0 µg/mL) against both mouse lymphomas (IC_50_ = 10 µg/mL) and against the three human cell lines, A549 (lung carcinoma), HT29 (colon carcinoma), and MEL28 (melanoma) human cell lines.

Pathogenic fungus *Gibberella zeae* (syn. *Fusarium roseum*) is a worldwide parasite that can produce a wide variety of steroids and synthesized unusual (22*E*,24*R*)-1(10→6)-abeoergosta-5,7,9,22-tetraen-3α-ol (**19**), which was isolated from the cultures of *Gibberella zeae*, an endophytic fungus isolated from the marine green alga *Codium fragile* [51]. This compound showed significant cytotoxicity toward murine colorectal CT26 and human leukemia K562 cancer cell lines. An aromatic B ring called topsentisterol E1 (**20**) was isolated from bioactive fractions of a marine sponge *Topsentia* sp. [52]. The presence of this unusual steroid may indicate that it is synthesized by the endophytic fungus, which is a symbiont in this sponge species.

### Comparison of Biological Activities of Natural Aromatic Steroids

It is known that the chemical structure of both natural and synthetic molecules predetermines biological activity, which makes it possible to analyze the structure–activity relationships (SAR). This idea was first proposed by Brown and Fraser more than 150 years ago, in 1868 [53], although, according to other sources, SAR originates from the field of toxicology, according to which Cros in 1863 determined the relationship between the toxicity of primary aliphatic alcohols and their solubility in water [54]. More than 30 years later, Richet in 1893 [55], Meyer in 1899 [56], and Overton in 1901 [57] separately found a linear correlation between lipophilicity and biological effects. By 1935, Hammett [58,59] presented a method of accounting for the effect of substituents on reaction mechanisms using an equation that considered two parameters, namely the substituent constant and the reaction constant. Complementing Hammett’s model, Taft proposed in 1956 an approach for separating the polar, steric, and resonance effects of substituents in aliphatic compounds [60]. Combining all previous developments, Hansch and Fujita laid out the mechanistic basis for the development of the QSAR method [61], and the linear Hansch equation and Hammett’s electronic constants are detailed in the book by Hansch and Leo published in 1995 [62].

Some well-known computer programs can, with some degree of reliability, estimate the pharmacological activity of organic molecules isolated from natural sources or synthesized compounds [63,64,65]. It is known that classical SAR methods are based on the analysis of (quantitative) structure–activity relationships for one or more biological activities using organic compounds belonging to the same chemical series as the training set [66].

The computer program PASS, which has been continuously updating and improving for the past thirty years [67], is based on the analysis of a heterogeneous training set included information about more than 1.3 million known biologically active compounds with data on ca. 10,000 biological activities [68,69]. Chemical descriptors implemented in PASS, which reflect the peculiarities of ligand–target interactions and original realization of the Bayesian approach for 18 elucidations of structure–activity relationships, provide the average accuracy and predictivity for several thousand biological activities equal to about 96% [70]. In several comparative studies, it was shown that PASS outperforms in predictivity some other recently developed methods for estimation of biological activity profiles [71,72]. Freely available via the Internet, the PASS Online web service [73] is used by more than thirty thousand researchers from almost a hundred countries to determine the most promising biological activities for both natural and synthetic compounds [74,75,76]. To reveal the hidden pharmacological potential of the natural substances, we have successfully been using PASS for the past fifteen years [77,78,79,80].

In the current study, we obtained PASS predictions for about one hundred steroids and triterpenoids produced by different living organisms. PASS estimates are presented as Pa values, which correspond to the probability of belonging to a class of “actives” for each predicted biological activity. The higher the Pa value is, the higher the confidence that the experiment will confirm the predicted biological activity [78,80].

The study of the biological activity of aromatic steroids using the PASS program showed that all steroids presented in Table 1 can be divided into two groups. The first group includes steroids that show antiovulation activity with a high degree of reliability, and the second group includes steroids for which antihypercholesterolemic activity is dominant. The first group includes aromatic steroids numbered **7**, **9**, **1**, **8**, and **10** with over 92.5 percent confidence, although the second group includes steroids **6**, **11**, **13**, **14**, **15**, and **19** with over 94.6 percent confidence, according to PASS.

A 3D graph of the predicted and calculated activities of aromatic steroids belonging to the first group is shown in Figure 2. From the first group of steroids that are inhibitors of ovulation, 2-hydroxyestrone (**7**) can be distinguished, which has a confidence level of more than 95%, and its predicted activities are presented in Figure 3. Additionally, 3D graphs that demonstrate the predicted pharmacological activity were obtained using the Origin Pro 2021 graphical program. The program analyzes the data obtained by the PASS program and builds graphs that are given in this publication.

Among the second group of aromatic steroids that demonstrate strong antihypercholesterolemic activity, and a 3D graph of predicted and calculated antihypercholesterolemic activity is shown in Figure 4; the strongest steroid is estrone glucuronide (**6**), and a 3D graph of its predicted pharmacological activities is shown in Figure 5.

## 3. Natural, Semi-, and Synthetic Steroid Phosphate Esters

Steroid phosphate esters are rare lipid molecules that can form the building blocks of biological membranes and have been found more recently in starfish extracts. So, for the first time, steroid phosphates were discovered more than a quarter of a century ago by Italian scientists from the University of Frederico II, in the city of Napoli. Unique steroids were isolated from the extract of the polar lipids of the starfish *Tremaster novaecaledoniae*, which was collected at a depth of 530 m of New Caledonia [110]. The isolated phosphated steroid glycosides were called tremasterols A and C, and their structures were identified as 3β-O-sulphated, 6α-O-phosphated, and 16β-O-acetylated groupings on a steroidal skeleton (**21**, **22**, and **23**).

Synthetic steroids, such as testosterone 17β-phosphate (**24**), cortisol 21-phosphate (**25**), and cholesterol 3β-phosphate (**26**), were chosen by us for comparison of biological activity with the activity of steroids isolated from marine invertebrates [111]. Testosterone 17β-phosphate (**24**) is an androgen and belongs to the class of anabolic steroids and is used for intramuscular injection, and it is a substrate for phosphatases in the phosphatase pool of the prostate [112]. Cortisol 21-phosphate (**25**) refers to the glucocorticoid class of hormones, and it functions to increase blood sugar levels through gluconeogenesis and to promote the metabolism of fats, proteins, and carbohydrates, and it is a substrate for alkaline phosphatase and is used for an enzyme immunoassay for human chorionic gonadotropin, human growth hormone, and α-fetoprotein and estradiol [113]. Cholesterol 3β-phosphate (**26**) promotes normalization of blood pressure and plays an important role in atherogenesis [114,115].

Two novel cholesterol-lowering agents called sodium ascorbic campestanol phosphate (**27**) and sodium ascorbic sitostanol phosphate (**28**) were synthesized, and their properties were studied [116]. Using Western blot analysis of P-GP expression, it was shown that changes in mdr-1 gene expression lead to correlating changes in P-GP protein expression. More recently, two steroid phosphate esters (**29** and **30**, structures see on Figure 6 and activities on Table 2) have been synthesized, acting as inhibitors of cholesterol biosynthesis. Methods of treating or preventing various diseases, conditions, and disorders by administering these steroids or compositions are also provided [117]. Estradiol phosphate (**31**) is ester of estrogen with phosphoric acid and acts as a prodrug of estradiol in the human body. In medical practice, both drugs can be used to treat prostate cancer [118].

### Comparison of Biological Activities of Steroid Phosphate Esters

According to PASS, antihypercholesterolemic activity is characteristic of all steroid phosphate esters presented in Table 2 with varying degrees of confidence. In addition, all steroids are inhibitors of cholesterol synthesis, except for steroid **31**. However, some lipid steroidal molecules such as **21**–**23** can be agents for wound healing and hepato-protectors with a high degree of certainty, and steroids **23**–**28** and **31** also demonstrate the properties of a neuroprotective agent with a high degree of certainty from 90 to 98%.

Steroid phosphate esters are of great practical interest in medicine, as they are wound healing agents. The confidence level of steroid 21 is 97.5%. Figure 7 shows the 3D graph of biological activities with a dominant property as a wound healing agent.

The sitostanol derivative (**30**) was synthesized by scientists from British Columbia (Canada) as a strong inhibitor of cholesterol synthesis, and as shown by the PASS data, this lipid molecule is in fact an excellent drug for the treatment of atherosclerosis. The wide range of biological activities of the sitostanol derivative (**30**) is shown in Figure 8.

## 4. Highly Oxygenated Natural Steroids and Triterpenoids

Highly oxygenated steroids are a large group of steroids and triterpenoids found in plant, fungal, and invertebrate extracts, and many of them exhibit a wide range of biological activities. This group of lipids includes secosteroids, epoxy steroids and peroxy steroids, and triterpenoids, which demonstrate a high degree of activity as potential regulators of lipid metabolism [119,120,121,122,123].

### 4.1. Secosteroids Derived from Marine and Terrestrial Sources

Secosteroids are a large group of natural steroidal hormones with a so-called ‘broken’ ring by oxidation of rings A, B, C, or D. Typical representatives of secosteroids are fat-soluble vitamins of group D [124,125,126,127]. Secosteroids are found in plant and animal extracts, produced by fungi, and found in marine invertebrates and algae [121,128,129,130].

Two secosterols, 3β-hydroxy-8α,9α-oxido-8,9-secoergosta-7,9(11),22-triene (**32**) and 3β-hydroxy-8α,9α-oxido-8,9-secoergosta-7,22-dien-12-one (**33**), named tylopiol A and B, respectively, were isolated from the fresh fruit bodies of fungus *Tylopilus plumbeoviolaceus* [131]. The other two steroids, named gloeophyllin J (**34**) and I (**35**), have been isolated from the solid cultures of North American wood-rotting fungi *Gloeophyllum abietinum*. Both compounds showed cytotoxic activity against human cancer cell lines K562 and HCT116 [132]. The chemical structures of steroids are shown in Figure 9 and the biological activity is shown in Table 3.

7-Oxasteroid (**36**) was isolated and characterized from the culture of *Aspergillus ochraceus* EN-31. This endophytic fungus is found and isolated from the marine brown alga *Sargassum kjellmanianum* (Dalian coastline, China) [133]. Compound **36** has been previously reported from a *Penicillium* sp. [134], and it displayed cytotoxic activity against NCI-H460, SMMC-7721, and SW1990 cell lines with IC_50_ values of 12.1, 16.9, and 67.6 μM, respectively [133].

Secosteroid (**37**) has been determined in extracts of Australian soft coral *Sinularia* sp. [135,136], and other octocoral *Sinularia leptoclados* are sources of bioactive 9,11-secosteroids and steroid 3,11-dihydroxy-9,11-secogorgost-5-en-9-one (**38**), which showed the highest peroxisome proliferator-activated receptor (PPAR) activity with an IC_50_ value of 8.3 μM for inhibiting human breast adenocarcinoma cell (MCF-7) growth. In addition, this steroid modulated the expression of various PPAR-regulated downstream biomarkers, including cyclin D1, cyclin-dependent kinase, B-cell lymphoma 2 (Bcl-2), p38, and extracellular-signal-regulated kinase [137,138].

Vitamin D2, known as ergocalciferol or 9,10-seco-(5Z,7E)-5,7,10(19),22-ergostatetraene-3β-ol (**39**), is the main one that is used in human nutrition [139]. All forms of vitamin D were found in mushrooms (brown Italian cremini, chanterelle, enoki, maitake, morel, shiitake, oyster, portobello, and white button mushrooms) and yeast [140,141,142,143,144,145]. Secosteroid, 3,11-dihydroxy-5,6-epoxy-9,11-secocholestan-9-one (**40**), was found and identified from extracts of the Taiwanese soft coral *Cespitularia taeniata* [146].

#### Comparison of Biological Activities of Secosteroids

Among the secosteroids shown in Figure 9, which show activity as regulators of lipid metabolism, the most interesting is lipid molecule number **40**, which has aAntihypercholesterolemic activity with a confidence level of 97.7 percent. Figure 10 shows the 3D graph of the predicted pharmacological activities of this steroid. Ergocalciferol (**39**) or vitamin D2 is also of great interest, as PASS showed strong antiparkinson activity with a confidence level of 96.0%. The full spectrum of the predicted pharmacological activities of ergocalciferol is shown in the 3D graph in Figure 11.

### 4.2. Natural Epoxy Steroids Derived from Marine Sources

α,α-epoxy- and/or β,β-steroids are found in lipid extracts of marine invertebrates, including sponges, soft corals, starfish, and nudibranchs (Mollusca) [123,155,156,157,158,159]. Two cytotoxic epoxysteroids, 5α,6α-epoxystigmasta-7,22-en-3β-ol (**41**) and 5α,6α-epoxystigmasta-7-en-3β-ol (**42**) were isolated from the ethanolic extract of the marine sponge *Ircinia aruensis* [160]. The chemical structures of steroids are shown in Figure 12 and the biological activity is shown in Table 4. Topsentisterol B4 (**43**), epoxy steroid with the β- and α-hydroxyl groups at position 3 and 7, respectively, were present in the extract of the far eastern sponge of *Topsentia* sp. [161]. (24E)-5α,6α-epoxystigmasta-7,24(28)-dien-3β-ol (**44**) was isolated from the South China Sea sponge *Phyllospongia foliascens* without studying the biological activity [162]. Triterpene glycoside, eryloside U (**45**) with be the 7,8-epoxide group was isolated from the sponge *Erylus goffrilleri* collected near Arresife-Seko Reef (Cuba) [163].

Unusual 17β,20β-epoxy-23,24-dimethylcholest-5-ene-3β,22-diol (**46**) was found in the Indian Ocean soft coral *Sarcophyton crassocaule* [164].

Two diol 22,23-epoxy steroids have been isolated from the marine sponge *Axinella* cf. *bidderi*, 17α-hydroxy-22β,23β-epoxycholest-5-en-3β-ol (**47**) and 17α-hydroxy-22β,23β-epoxy-24-methylcholest-5-en-3β-ol (**48**). Isolated steroids showed activity against prostate, ovary, pancreas, colon, and lung cell lines in vitro [165].

#### Comparison of Biological Activities of α,β-Epoxy Steroids Derived from Marine Sources

Comparing the data from the PASS obtained for the α, β-epoxy steroids presented in Table 4, it can be concluded that there are no outstanding steroids with lipid metabolism regulator properties; then, we present in Figure 13 a comparative 3D graph for this subgroup of steroids.

For a comparative graphic characterization of α, β-epoxy steroids, we selected steroids numbered **42**, **44**, and **48**. All these lipids are characterized by the property of regulators of lipid metabolism with dominant antihypercholesterolemic activity. According to PASS data, steroids **42** and **44** also show a high level of anticancer activity, while steroid **48** is additionally characterized by neuroprotective properties.

### 4.3. Peroxy-Type Steroids Derived from Natural Sources

Natural and/or synthetic compounds containing a peroxy group (R-O-O-R) are called peroxides [166,167,168,169,170,171,172]. Natural peroxides represent a rather large group of compounds that many microorganisms produce, and they have also been found in plants, mushrooms, animals, and marine invertebrates [166,167,168]. Peroxy steroids are a small group of natural lipids, mainly found in leaves, roots, and bark of plants, and are produced by fungal endophytes and are found in mushrooms [166,167,173].

#### 4.3.1. Steroid Endoperoxides

*Astropecten polyacanthus* starfish extract has significant cytotoxic effects and contains an unusual peroxy steroid called astropectenol B (**49**) was isolated from a methanol extract of this starfish [174].

Cytotoxic steroid, (3,5,8,24R,25R)-epidioxy-24,26-cyclocholesta-6,9(11)-dien-3-ol (**50**) was identified from marine sponge *Tethya* sp. [175]. (3,5,8,24R)-Epidioxy-24-methylcholest-6-en-3-ol (**51**) was detected in MeOH extract of the marine sponge *Luffariella* cf. *variabilis* [176], and 22,23-dihydro-5,8-epidioxystigmast-6-en-3-ol (**52**) was found in the marine sponges *Luffariella* cf. *variabilis* and *Tethya* sp. and sea squirt *Dendrodoa grossularia* [175,176,177,178].

Fuscoporianol D (**53**) was found in field-grown mycelia of fungus *Inonotus obliquus* (family Hymenochaetaceae) [179]. Ergosterol peroxide 3-O-β-D-glucopyranoside (**54**) produced by fungus *Tremella fuciformis* [180] and same compound was detected in the fruiting bodies of the Chinese toxic woodland mushroom *Naematoloma fasciculare* [181]. Endoperoxy steroid **55** was found in popular mushroom in Japan, *Buna shimeji* and in oyster fungus *Pleurotus ostreatus* [182]. Two endoperoxy glycosides (**55** and **56**) were found in ethanol extracts of the fungus *Lactarius volemus*, which demonstrated anticancer activity [183,184]. The chemical structures of steroids are shown in Figure 14, and the biological activity is shown in Table 5.

#### 4.3.2. Steroid and Triterpenoid Hydroperoxides

A flowering herbaceous perennial plant from the family Araceae, Arum italicum, also known as Italian arum and Italian lords-and-ladies, contains a suite of hydroperoxysterols, including two (**57** and **58**) that are interesting for lipid metabolism [185]. Two steroids (**59** and **60**), which showed a cytotoxic effect against several human cancer cell lines, have been isolated from the bark of the chinaberry tree, *Melia azedarach* [186]. The chemical structures of steroids are shown in Figure 15, and the biological activity is shown in Table 6.

Ponce and co-workers obtained ergosterol 7-hydroperoxide (**61**) by the photo-oxidation of ergosterol with singlet oxygen in vivo and in vitro [187]. In addition, the yeast *Saccharomyces cerevisiae* with the singlet oxygen leads to rapid oxidation of ergosterol to ergosterol 7-hydroperoxide (**61**) [188].

A tree called Sakae Naa (*Combretum quadrangulare*) in Vietnam, Cambodia, Laos, Myanmar, and Thailand contained a cycloartane-type triterpene quadrangularic acid F (**62**), and the aqueous and EtOH extracts show antibacterial, anti-HIV, hepatoprotective, and cytotoxic activities [189,190,191].

The plant *Proboscidea louisiana* produced dammarane triterpenes known as probosciderol I (**63**) [192], and the stem bark of *Rhus javanica* contained isofouquierone peroxide (**64**) [193]. The leaves of *Melaleuca ericifolia* contained antiproliferative norlupane triterpene (**65**) [194], and the aerial roots of *Ficus microcarpa* afforded similar norlupane triterpene (**66**) [195].

#### 4.3.3. Comparison of Biological Activities of Peroxy Steroids Derived from Natural Sources

According to published data, most natural peroxides isolated from both plants and marine invertebrates show predominantly antiprotozoal activity. Such compounds include diterpenoids, triterpenoids, and steroids [166,167,168,171,173,196,197,198,199].

Analysis of PASS data on peroxy steroids and triterpenoids such as endoperoxides and hydroperoxides showed that most of these lipids have a high confidence level of more than 90 percent, but nevertheless, only three of all peroxy steroids deserve attention; these are steroids numbered **59**, **63**, and **66**, which have a confidence level of over 93 percent. Figure 16 presents the 3D graph showing the predicted and calculated biological activity of steroid hydroperoxides.

## 5. Carbon-Bridged Steroids (CBS) and Triterpenoids

In both natural and synthetic steroids, when an additional ring is formed within the steroid skeleton, through a direct bond between any two carbon atoms (or more) of the steroid ring system or an attached side chain, such steroids (or triterpenoids) are called carbon-bridged steroids [13,200,201,202].

Of the more than 500 carbon-bridged steroids and triterpenoids studied, we found only twelve lipids that have a confidence level of more than 90 percent as potential lipid regulators. We give their description and their sources in nature below [13,203].

Studying the photoproducts obtained by photochemical processes of vitamin D, the cyclobutane containing derivative **67** was identified [204], and similar secosteroid named toxisterol (**68**), as a minor transformation product of vitamin D2, has been found in various mushrooms [205]. The chemical structures of steroids and triterpenoids are shown in Figure 17, and the biological activity is shown in Table 7.

A unique steroid containing a 5,19-cycloergostane skeleton, (3β,5β,6β,7α,22*E*,24ς)-5,19-cycloergost-22-ene-3,6,7-triol, named hatomasterol (**69**), was found in the extracts of the Okinawan sponge *Stylissa* sp., and this compound demonstrated cytotoxicity against HeLa cells in vitro [206]. Steroidal saponins named poecillastrosides E (**70**) and G (**71**), an oxidized methyl at C-18, into a primary alcohol or a carboxylic acid, have been found in extracts of the Mediterranean deep-sea sponge *Poecillastra compressa*. Poecillastroside E bearing a carboxylic acid at C-18 showed antifungal activity against *Aspergillus fumigatus* [207], and other cyclopropyl containing steroids (**72**) and (**75**) were found in the methanol extract of the marine sponge *Petrosia weinbergi* [208]. Cycloart-24-en-3-ol (**73**) was detected in ethanol extract of marine green alga *Cladophora fascicularis* [209].

A cytotoxic sterol named petrosterol (**74**) showed cytotoxic activities on A549, HL-60, MCF-7, SK-OV-3, and U937 cancer cell lines, and was present in extracts of several marine sponges such as Vietnamese sponge *Ianthella* sp., *Petrosia spheroïda* from the Indian Ocean, *Halichondria* cf. *panicea* of the Japanese island Iriomote, and Japanese marine sponge *Strongylophora corticate* [210,211,212,213].

A rare steroid named calysterol (**76**), the minor sterol component of the sponge *Calyx niceaensis* and *Petrosia ficiformis*, possessing the unique feature of a cyclopropene ring bridging C23,24, and isocalysterol (**77**), was detected in the same sponge [214,215,216,217,218]. Sterol ester, 24,26-cyclo-5α-cholest-(22*E*)-en-3β-4′,8′12′-trimetyltridecanoate (**78**), has been isolated from a deep-water marine sponge, *Xestospongia* sp. [218].

### Comparison of Biological Activities of CBS Steroids and Triterpenoids

Carbon-bridged steroids (CBS) and triterpenoids belong to a rare group of natural hormones found in various natural sources such as green, yellow-green, and red algae, sea sponges, soft corals, ascidians, starfish, and other marine invertebrates. In addition, this group of rare lipids is found in amoebas, fungi, fungal endophytes, and plants [13,203].

We have isolated carbon-bridged steroids presented in Figure 18, which, according to the PASS data, have a confidence level of more than 90 percent. Among this group of lipids, we identified three, numbered **70**, **74**, and **78**, the activity of which most clearly reflects their regulatory functions of lipid metabolism, with antihypercholesterolemic properties dominating. In addition, these steroids can be used as inhibitors of cholesterol synthesis and as drugs for the treatment of atherosclerosis. The 3D graph demonstrating the predicted and calculated biological activity of carbon-bridged steroids is shown in Figure 18.

## 6. Neo Steroids Derived from Terrestrial and Marine Sources

Secondary metabolites containing a tertiary butyl group (or tertbutyl unit) are rather rare compounds found in cyanobacteria, plant leaves, fungi, marine invertebrates, and algae [12,219,220,221].

Neo steroids are a small group of lipids that are synthesized by yeast and fungi and are found in various parts of plants. In recent years, with the improvement of steroid analysis methods, they have been found in seaweeds, marine sponges (class Demospongiae), anemones (class Anthozoa), and cucumbers (class Holothuroidea) [12,219].

Two neo steroids (**79** and **85**) were present in leaves and stems and the pericarp of the fruit and roots of a plant from the family Cucurbitaceae [222], and another sterol, 24-methylene-25-methyl-lathosterol (**80**), was isolated from aerial parts of the herbaceous plant, *Sicyos angulatus* [223]. The chemical structures of steroids and triterpenoids are shown in Figure 19, and the biological activity is shown in Table 8. 

Aerial parts of the strongly aromatic herb *Ocimum basilicum* from the Labiatae family, such as seeds, flowers, and roots, are widely used as medicines [224,225,226]. The leaves and flowers of this plant are used in folk medicine as a tonic and anthelmintic [227]. Leaf tea is used to treat flatulence and dysentery, while the plant’s oil may be useful for relieving mental fatigue, colds, cramps, rhinitis, and as a first aid for treating wasp stings and snake bites [227,228]. A species of this plant from Pakistan contains (22E)-24ξ-ethyl-25-methylcholesta-5,22-diene-3β-ol-3-O-D-gluco-pyranoside (**81**) [229].

Three neo steroids (**82**, **83** and **86**) have been isolated from this auxotroph mutant *Saccharomyces cerevisiae* strain GL7 using appropriate substrates for biosynthesis [230]. The sterol C24-methyl transferase from *Trypanosoma brucei* TbSMT1 produces 24-methyl sterols that serve as substrates for 24-dimethyl sterols that contain a Δ25(27)-bond, and the neo steroid (**84**) was isolated from the extract *S. cerevisiae* [231]. Topsentinols C (**87**) and E (**88**) contain a tertiary butyl group in steroids, and these neo steroids were obtained from the Okinawan marine sponge *Topsentia* sp. [232,233]. (3β,24E)-25-Methylstigmasta-5,24(28)-dien-3-ol (**89**) and axinyssasterol (**90**) were identified from a marine sponge *Pseudoaxinyssa* sp. [234].

Neo steroid (3β,22E,24ξ)-28,28-dimethyl-stigmasta-5,22,25-trien-3-ol acetate (**91**) and (3β,22E)-25-methyl-stigmasta-5,22-dien-3-ol (**92**) were obtained from an ethanol extract of the sponge *Halichondria* sp. [235,236], and the compound (**93**) was also found in specimens of the sponge *T. aplysinoides* from the inshore waters of Sri Lanka [236]. The antimicrobial halistanol (**93**) was isolated from the Okinawan sponge *Halichondria* cf. *moorei* more than 25 years ago [237].

The Atlantic tropical sponge *Erylus goffrilleri* contains unusual lanostane glycosides, erylosides R (**94**) and T (**95**), and the same glycosides were isolated from the Caribbean sponge *E. goffrilleri*. Both eryloside glycosides R and T exhibit cytotoxic activities against Ehrlich carcinoma tumor cells [238,239].

### Comparison of Biological Activities of Neo Steroids

Neo steroids are a rare group of naturally occurring lipid molecules that exhibit high levels and a wide range of activities. The chemical structures shown in Figure 19 have been found in plant and marine invertebrate extracts. According to PASS data, neo steroids show a high confidence level of up to 98.9 percent, with antihypercholesterolemic activity being dominant. In addition, virtually all neo steroids are cholesterol synthesis inhibitors and can be used to treat atherosclerosis and related diseases.

Of the seventeen neo steroids, we have selected four that show a high confidence level of 96.9 to 98.9 percent. Figure 20 demonstrates the 3D graph and shows the predicted and calculated biological activity of neo steroids (compound numbers: **81**, **86**, **89,** and **93**) showing the highest degree of confidence, more than 96.9%.

## 7. Miscellaneous Steroids and Triterpenoids Derived from Marine Sources

In this section, we have collected steroids that do not belong to the first six groups of triterpenoids but show high antihypercholesterolemic activity. Mostly, the steroids shown in Figure 20 are found in soft corals collected in various regions of the world’s oceans. An interesting question is why from coral? The fact is that, by studying the activities of various marine organisms, we concluded that corals or their fungal endophytes synthesize many biologically active metabolites. It is well known that corals are associated with many microscopic fungi, so a reasonable question arises: what synthesizes bioactive molecules in corals? Whether they do so themselves or their fungal endophytes or bacteria is a question that remains open [240,241,242].

Cytotoxic steroid called stereonsteroid G (**96**) was isolated from the methylene chloride extract of the Formosan soft coral *Stereonephthya crystalliana*. The extract of this coral showed significant cytotoxicity against A549, HT-29 and P-388 cancer cells in vitro [243]. The chemical structures of steroids and triterpenoids are shown in Figure 21, and the biological activity is shown in Table 9.

Trihydroxy sterol, pregna-5-ene-3,20,21-triol (**97**), has been isolated from the Gulf of California gorgonian *Muricea* cf. *austera* [244], and a rare spiroketal steroid, 22-acetoxy-3,25-dihydroxy-16,24,20–24-bisepoxy-(3,16,20S,22R,24S)-cholest-5-ene (**98**) was found in extracts of the Indian Ocean gorgonian, *Gorgonella umbraculum* [245]. The cholestane class steroidal hemiacetals named anastomosacetal A (**99**) was obtained from the gorgonian coral *Euplexaura anastomosans*, collected off the shore of Keomun Island, South Sea Korea [246], and petasitosterone B (**100**) was isolated from a Formosan marine soft coral *Umbellulifera petasites* [247]. Petasitosterone B (**100**) displayed inhibitory activity against the proliferation of a limited panel of cancer MOLT-4 and DLD-1 cell line. Steroidal glycoside (**101**) was isolated from water–methanol solutions of the soft coral *Sinularia gibberosa* [248], and steroid **102** was found in the methanol extract of the Vietnamese soft coral *Sinularia nanolobata* [249]. The minor sterol **103** was isolated from the soft coral extract of the genus Sinularia, and it was synthesized [250]. Steroid named crassarosteroside A (**104**) was obtained from *Sinularia granosa* and *S. crassa* soft coral extracts [251,252], compound **105** has been isolated from *Sinularia conferta* and *S. nanolobata* [253,254], and metabolite **106** was detected in MeOH extract of the soft coral *S. cruciata* [255]. Oxysterol (**107**) was detected in extracts of octocoral of the genus Gorgonia from the eastern Pacific, Panama [256].

### Comparison of Biological Activities of Soft Coral Steroids

The chemical and structural diversity of soft corals sterols and triterpenoids is well known and has been documented in several review articles in the literature [128,130,257,258,259,260,261]. PASS analysis shows that the steroids in Figure 21 synthesized by soft corals are indeed of interest due to their high anticholesterol activity. Both the biological activities found using PASS and those obtained experimentally coincide. In addition, we have identified three steroids, numbered **99**, **101**, and **104**, which demonstrate antihypercholesterolemic activity with a high degree of certainty. Figure 22 shows a 3D graph of predicted and calculated biological activity of these steroids.

## 8. Synthetic and Semi-Synthetic Steroids and Triterpenoids and Comparison of Their Biological Activities

Steroid hormones and triterpenoids belong to the group of physiologically active substances (sex hormones, corticosteroids, etc.) that regulate vital processes in vertebrates and many species of invertebrates and humans [262,263,264,265]. They are regulators of the fundamental vital processes of a multicellular organism-coordinated growth, differentiation, reproduction, adaptation, and behavior [266,267,268]. For more than 90 years, steroid hormones and triterpenoids have been the subject of close attention of chemists since synthetic analogues have long replaced natural steroids and triterpenoids [269,270,271,272,273]. It is known that semi-synthetic or synthetic steroid hormones have properties that steroids do not have, isolated from natural sources. In this section, we present synthetic and semi-synthetic steroids and triterpenoids that contain heteroatoms and do not exist in nature but demonstrate activities that are necessary for pharmacologists and physicians.

### 8.1. Bioactive Epithio Steroids and Triterpenoids

Anabolic steroids are pharmacological drugs that mimic the effect of the male sex hormone testosterone and its derivatives [274,275,276]. Anabolic steroids accelerate the synthesis of protein within cells, which leads to a pronounced hypertrophy of the muscle tissue, because of which, they have found wide application in sports medicine and bodybuilding [277,278,279].

Semi- and/or synthetic epithio steroids represent a rare group of bioactive lipids, since they are hydrophobic molecules insoluble in water, which are not found in nature. Epithio steroids have been reported to possess a variety of cytotoxic activities, and they are widely used as anticancer agents. The thiirane group is an important substance and shows some promising biological activities. Steroids containing an epithio group in positions 2 and 3 are anabolic steroids and are widely known and used in sports medicine. Representatives of this group of steroids are of great interest for pharmaceutical chemistry and medicine [280,281,282,283,284,285,286]. The most widely known are such epithio steroids that are used in sports pharmacology and medicine: epistane (**108**, 2α,3α-epithio-17α-methyl-5α-androstan-17β-ol), epitiostanol (**109**, 2α,3α-epithio-5α-androstan-17β-ol), a known potent antiestrogenic and antitumor agent, and hemapolin (**110**, 2β,3β-epithio-17α-methyl-5α-androstan-17β-ol) [282,283,287,288,289,290,291]. The chemical structures of epithio steroids are shown in Figure 23 and the biological activities are shown in Table 10. Presented in Figure 24, the anabolic 2α,3α-epithio chlostane (**111**) shown dominant anticancer activity, and steroid (**112**) demonstrated dominant properties as an antisecretory agent with 96.7% confidence and acts as an estrogen antagonist with 94.6% confidence.

Epithio steroids (**113**–**115** and **117**) and are cholesterol antagonists, and the anticancer triterpenoid (**116**), which was synthesized from a natural sample of 18β-glycyrrhetinic acid [292,293], shows lipid metabolism regulator properties.

### 8.2. Bioactive Selena Steroids

Selenium is an essential metalloid, and it is one of the most necessary trace elements for humans [296]. Selenium occupies an important place in the regulation of metabolism in humans, and therefore, it is necessary to monitor its presence in consumed foods [297,298]. The Allium and Brassica families as well as Brazil nuts, mushrooms (shiitake and white mushrooms), beans, chia seeds, brown rice, sunflower, sesame and flax seeds, and cabbage and spinach contain high enough selenium and organoselenium concentrations [299,300].

There are also many excellent reviews in the literature, which are devoted to the biological role and functions of organoselenium compounds [301,302,303,304]. Apparently, selena steroids are the main group of the essential metalloids that have been synthesized over the past 50 years, and approximately 300 have been synthesized [304,305,306,307,308,309].

The selena steroids numbered **118**, **119**, **124**, **125**, **126**, and **127** show dominant antihypercholesterolemic activity with a low degree of confidence from 90.5 to 91.2%, although for steroid number **128**, the confidence was 95.3%. For another group, the selena steroids numbered **120**, **121**, **122**, and **123**, the dominant properties are hyperlipemia, treatment of atherosclerosis, and treatment of lipoprotein disorders with a strong degree of confidence up to 99.6%. The chemical structures of steroids are shown in Figure 25, and the biological activity is shown in Table 11. Figure 26 shows the 3D graph the predicted and calculated biological activity of the selena steroids numbered **120**, **121**, and **122** with dominant properties as hyperlipemia and treatment of atherosclerosis.

### 8.3. Bioactive Tellura Steroids

Tellura steroids are a rare group of organic synthetic compounds whose biological activity is of great interest for medicine, pharmacology, and the pharmaceutical industry [304,306,308,309,311]. The chemical structures of steroids and triterpenoids are shown in Figure 27, and the biological activity is shown in Table 12.

We found only four tellura steroids, which exhibit properties as regulators of lipid metabolism, dominated by antihypercholesterolemic activity. However, the most interesting from the point of view of pharmacological values is tellura steroid with number **129**. In addition to its antihypercholesterolemic activity, it is worth pointing out that this steroid has also been shown to be used as an agent for the treatment of neurodegenerative diseases Alzheimer’s and Parkinson’s with strong confidence, over 94 percent. The 3D graph demonstrating the predicted and calculated biological activity of the tellura steroid (**129**) is shown in Figure 28.

### 8.4. Bioactive Astatosteroids

Astatine is natural radioelement that has short-lived isotopes, and synthetic organic astatine compounds are commonly used for radiotherapy [312,313,314]. Steroids containing astatine, which are called astatosteroids, were first synthesized approximately 40 years ago [315]. Some astatosteroids (2- and 4-astatoestradiol and 6-At-cholesterol, **135**, **136**, and **137**) have been synthesized in high radiochemical yields by the reaction of ^211^At/I_2_ and the corresponding chloromercury compounds. The stability in vitro was determined under different conditions in comparison with the analogous iodo compounds [313]. More recently, 6-astatomethyl-19-norcholest-5(10)-en-3β-ol (**134**) was synthesized at a yield of 60–70% [316]. The biological activity of these compounds has not been determined. The predicted biological activity of astatosteroids is presented in Table 13. The most characteristic biological properties for these steroids were antineoplastic, antiseborrheic, antisecretoric, and antihypercholesterolemic activities. The chemical structures of steroids are shown in Figure 29 and the biological activity is shown in Table 13. For all astatosteroids shown in Figure 29, antihypercholesterolemic activity is dominated. In addition, all steroids of this group, as shown by PASS, have properties as a treatment for bone diseases. This is a rare property for steroids. Figure 30 shows the 3D graph the predicted and calculated biological activity of the astatosteroid (**133**).

## 9. Conclusions

This review focuses on the intriguing topic of lipid metabolism regulation. The literature does not fully describe the means that regulate lipid metabolism. Steroids and triterpenoids presented in this review are of great interest for medicine, and some of them may be potential regulators of lipid metabolism. However, experimental work is required to confirm this thesis. In the world in general, and in North America in particular, the study of biological activities using computer programs is gaining popularity. This is due to the fact that the number of isolated natural and synthetic compounds has long exceeded 20 million; there is no technical possibility of determining biological activity experimentally. Using the PASS program for the last fifteen years has shown that we are on the right track. During this time, we have investigated over 10,000 compounds and identified their potential biological activities. Based on early studies, we have selected a group of steroids and triterpenoids that are presented in this review and correspond to the name of the topic, as potential regulators of lipid metabolism.

## Figures and Tables

**Figure 1 marinedrugs-19-00650-f001:**
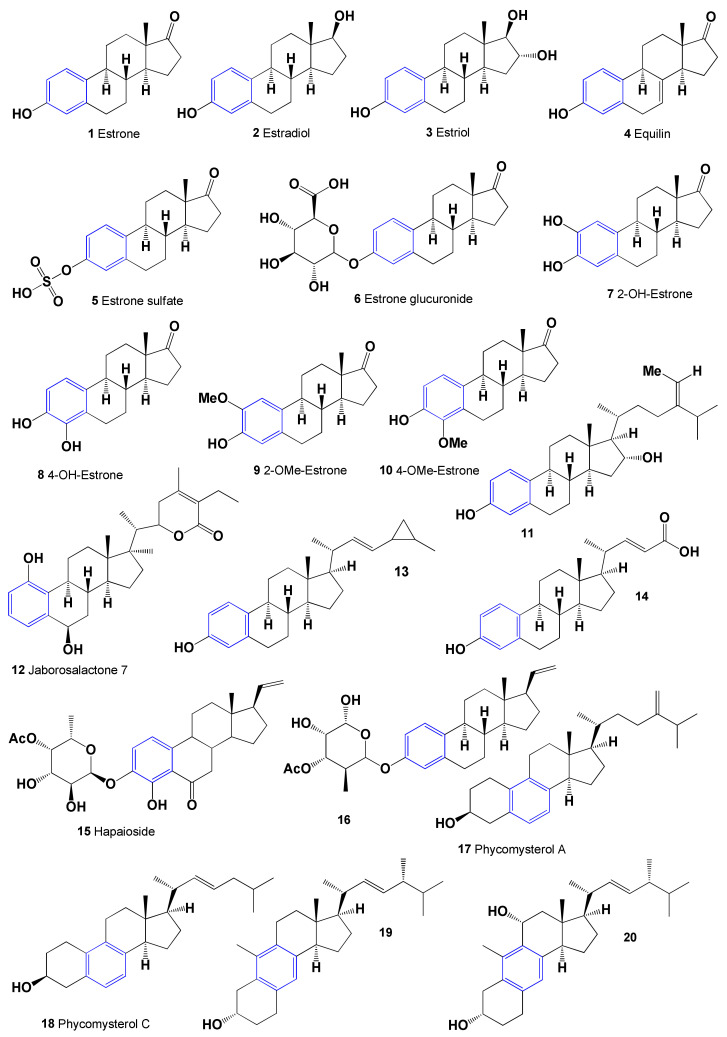
Bioactive aromatic steroids demonstrating with dominance of antihypercholesterolemia and antiovulation activity.

**Figure 2 marinedrugs-19-00650-f002:**
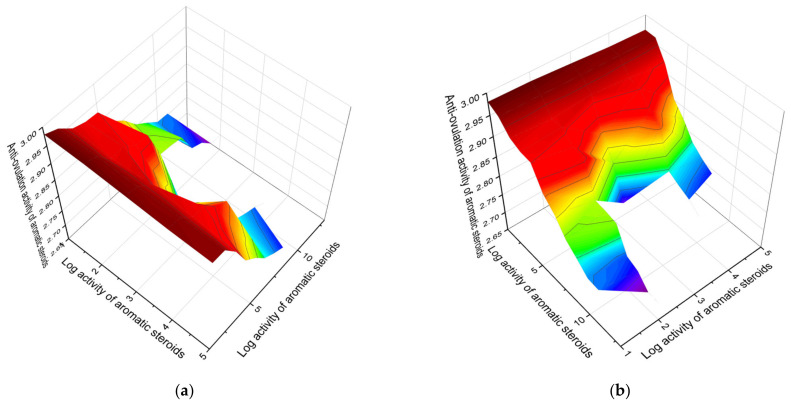
The 3D graph (X (**a**) and Y (**b**) views) shows the predicted and calculated antiovulation activity of aromatic steroids (compound numbers: **7**, **9**, **1**, **8**, and **10**) showing the highest degree of confidence, more than 92.5%. These steroids are derived from animals, including humans, as well as the extract of bark from the main wooden rod of ketapang *Terminalia catappa*, and can be used in clinical medicine as potential agents with strong ovulation inhibitors. The units of measurement of the x–y digits (Cos–Sin or Sin–Sin) are the relative dimensions that the Origin Pro 2021 graphics program chooses independently, depending on the data obtained by the PASS program. On the 3D graphs, the Origin Pro software in red indicates the maximum biological activity of an individual steroid, and blue indicates the minimum activity.

**Figure 3 marinedrugs-19-00650-f003:**
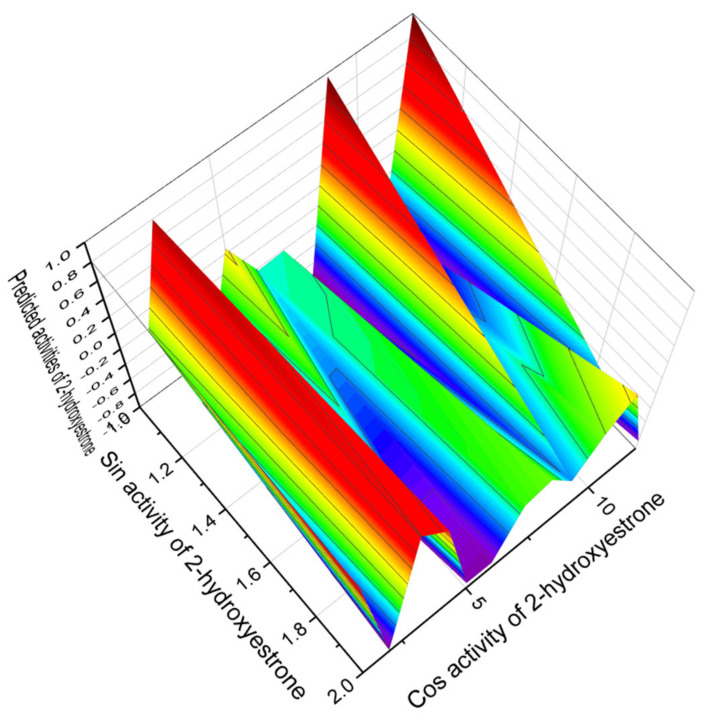
The 3D graph shows the predicted pharmacological activities of 2-hydroxyestrone (**7**). 2-Hydroxyestrone (2-OHE1) or 2.3-estracatechol, also known as estra-1,3,5(10)-trien-2,3-diol-17-one, is an endogenous, naturally occurring catechol estrogen that was first isolated from human urine more than 60 years ago [106], apparently as a product of estrone and estradiol metabolism. According to the PASS data, 2-hydroxyestrone is the strongest ovulation inhibitor among aromatic steroids. In addition, it is a cardiovascular analeptic and can be used as a drug for the treatment and prevention of male reproductive dysfunction, menopausal disorders in women, as well as treatment of muscular dystrophy.

**Figure 4 marinedrugs-19-00650-f004:**
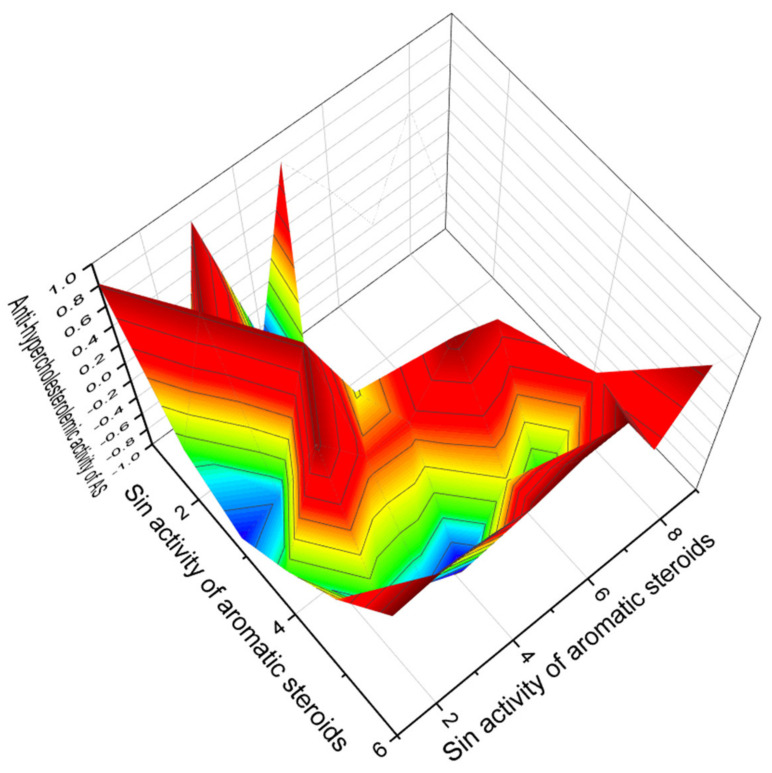
The 3D graph showing the predicted and calculated antihypercholesterolemic activity of aromatic steroids (compound numbers: **6**, **13**, **11**, **14**, **15**, and **19**) with the highest degree of confidence, more than 94.6%. These steroids have been obtained from a variety of natural sources, including plant, fungal, and marine invertebrate extracts, and appear to be useful in clinical medicine as potential agents with potent antihypercholesterolemic activity.

**Figure 5 marinedrugs-19-00650-f005:**
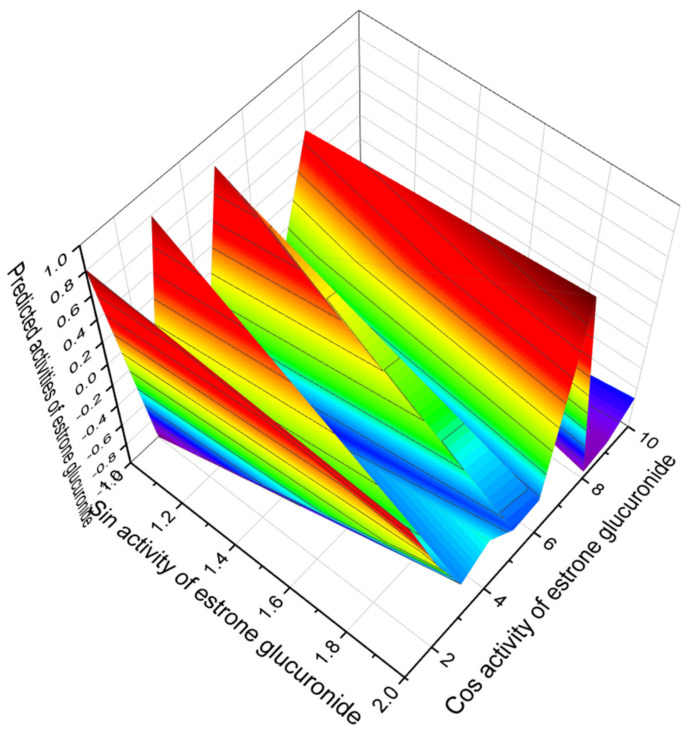
The 3D graph shows the predicted pharmacological activities of estriol 3-glucuronide (**6**). The water-soluble conjugate of estriol, also known as estriol 3-β-D-glucosiduronic acid, is a naturally occurring steroidal estrogen containing β-D-glucopyranuronic acid that was first discovered and isolated from the urine of women in the late 1930s [107,108,109]. According to the PASS data, estrone glucuronide, in addition to the main antihypercholesterolemic activity (97.3%), is also a potent regulator of lipid metabolism with a reliability of 90.7% and can be used to treat acute neurological disorders (92.2%).

**Figure 6 marinedrugs-19-00650-f006:**
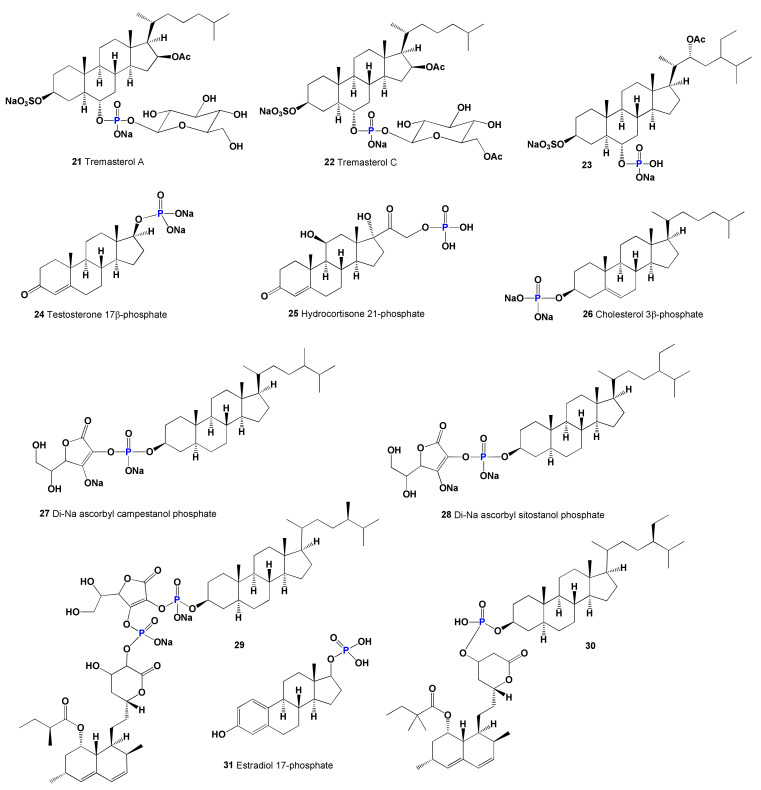
Bioactive natural, semi-synthetic, and synthetic steroid phosphate esters.

**Figure 7 marinedrugs-19-00650-f007:**
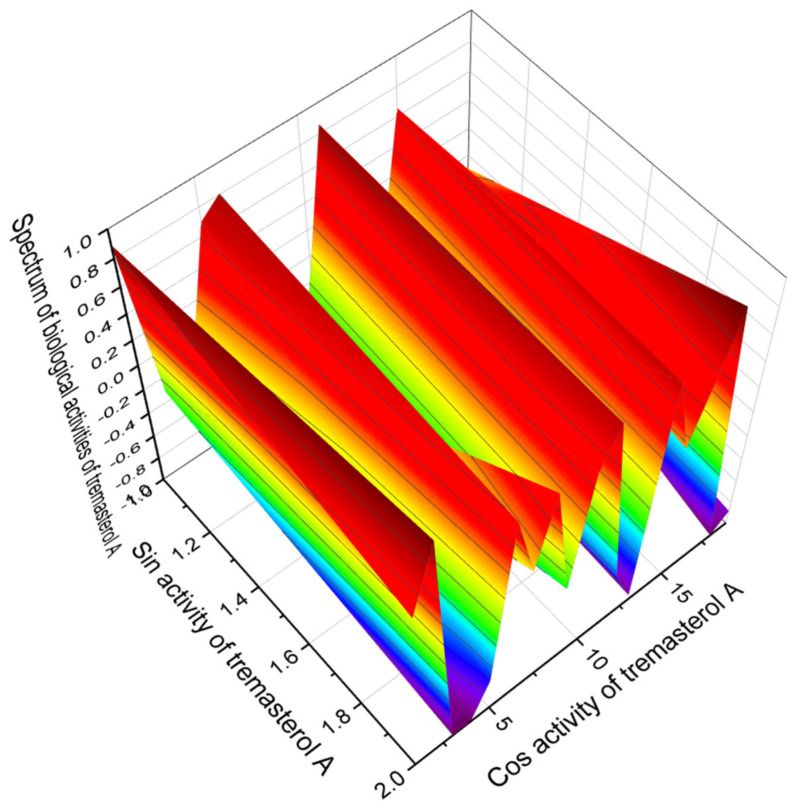
The 3D graph showing the predicted pharmacological activities of tremasterol A (**21**). Data from the PASS program show that phosphated steroid glycoside named tremasterol A exhibits 20 different biological activities, with dominant properties as a wound healing agent. Interestingly, tremasterol A is isolated from the starfish *Trenzaster novaecaledoniae.* This is a rare occasion when glycosides from starfish demonstrate such beneficial biological activities.

**Figure 8 marinedrugs-19-00650-f008:**
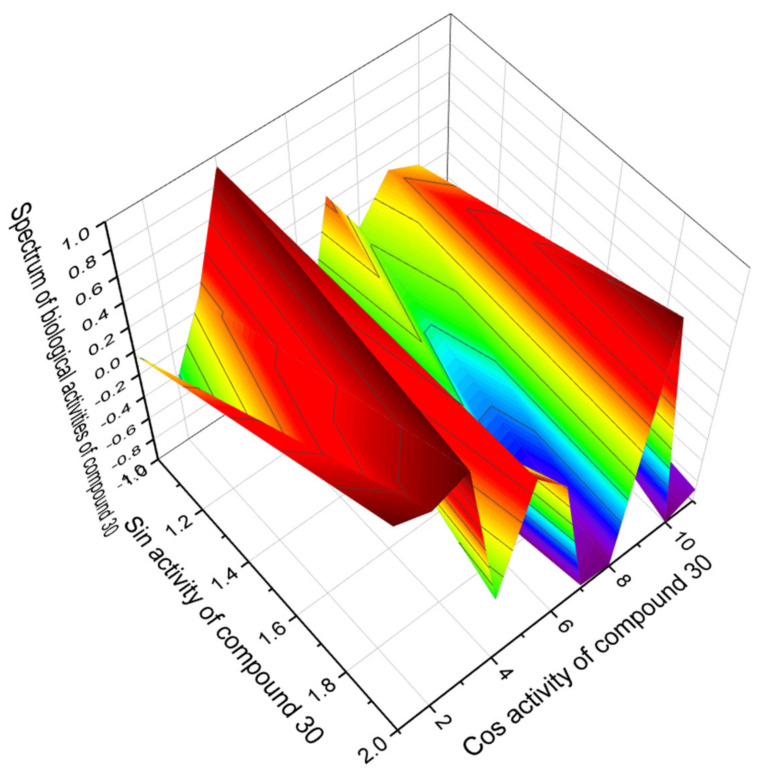
The 3D graph showing the predicted pharmacological activities of the sitostanol derivative (**30**). This drug has antihypercholesterolemic and antihyperlipoproteinemic properties that help lower cholesterol levels in the human body and reduce the risk of strokes and heart attacks. In addition, the drug is a strong cholesterol absorption inhibitor with a confidence level of 95.7% and a regulator of lipid metabolism with a confidence level of 95.4% and can be used to treat acute neurological disorders.

**Figure 9 marinedrugs-19-00650-f009:**
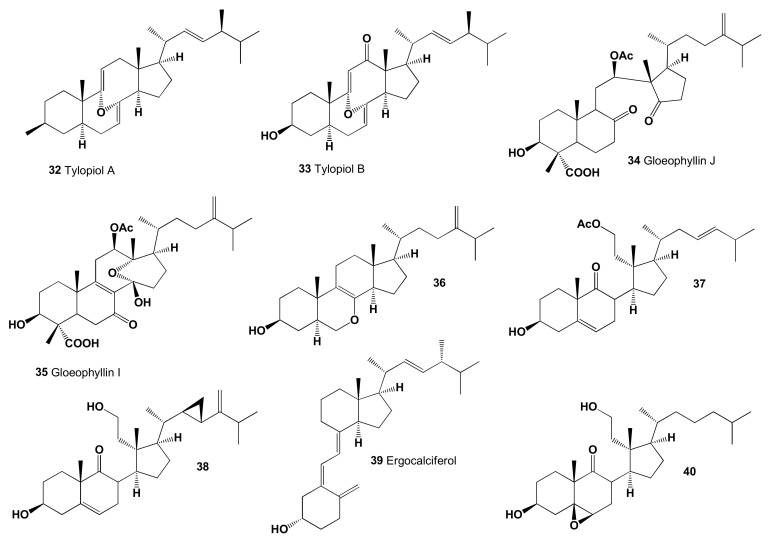
Secosteroids showing activity as lipid metabolism regulators.

**Figure 10 marinedrugs-19-00650-f010:**
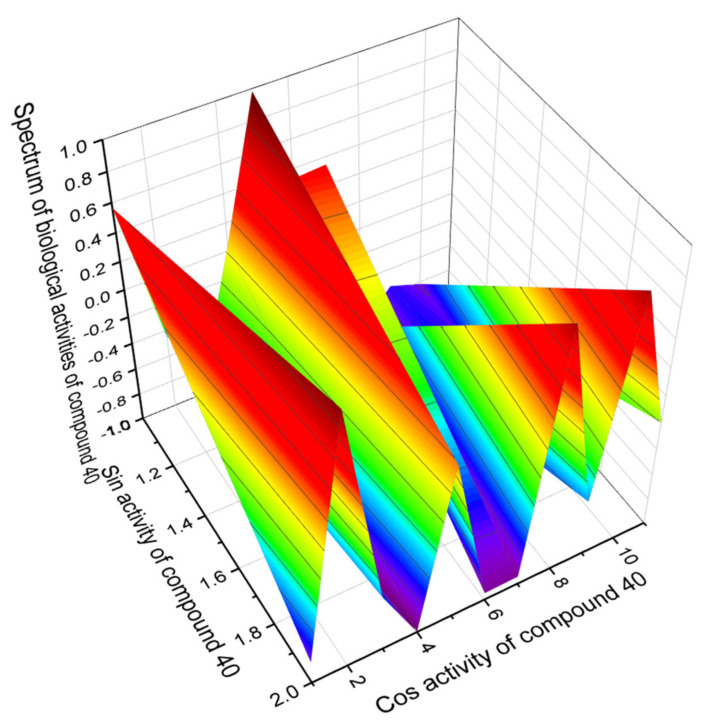
The 3D graph shows the predicted pharmacological activities of compound (**40**). This secosteroid is characterized by antihypercholesterolemic properties. In addition, it exhibits hepatoprotective properties, and it is an inhibitor of cell proliferation, which de facto can prevent several pathological diseases such as atherosclerosis, rheumatoid arthritis, psoriasis, idiopathic pulmonary fibrosis, scleroderma, and liver cirrhosis.

**Figure 11 marinedrugs-19-00650-f011:**
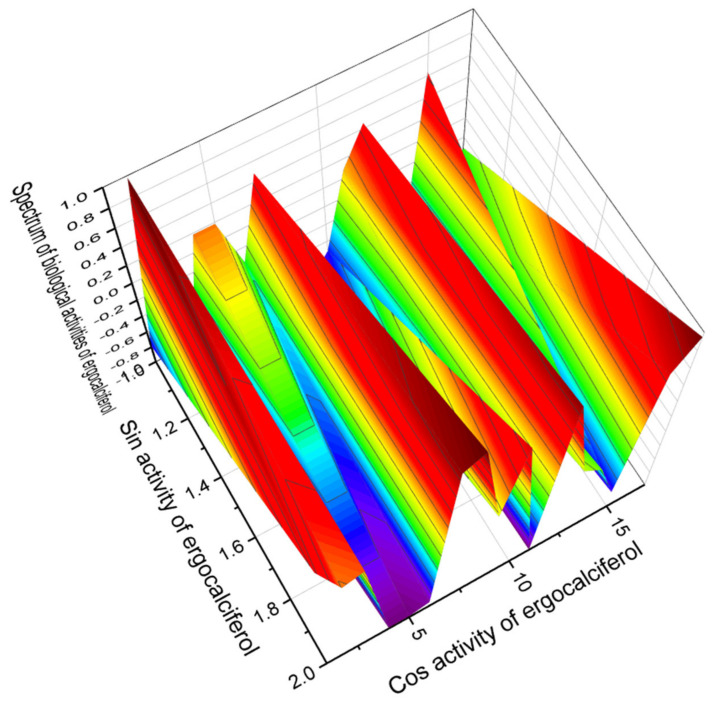
The 3D graph showing the predicted pharmacological activities of ergocalciferol (**39**). A secosterol hormone called ergocalciferol was first isolated as a radiation product of 7-dehydrocholesterol and described in 1936 by Windaus and co-workers [149]. Ergocalciferol is an important drug and is recommended by the World Health Organization. In addition, it has anticancer, anti-inflammatory, and antieczema properties, and can also be recommended as an antihypercholesterolemic agent and an agent against Parkinson’s disease. Certain foods, such as breakfast cereals and margarine, contain ergocalciferol in some countries, and it is found in the lichen *Cladina arbuscula* and alfalfa (*Medicago sativa*) [150,151,152,153,154].

**Figure 12 marinedrugs-19-00650-f012:**
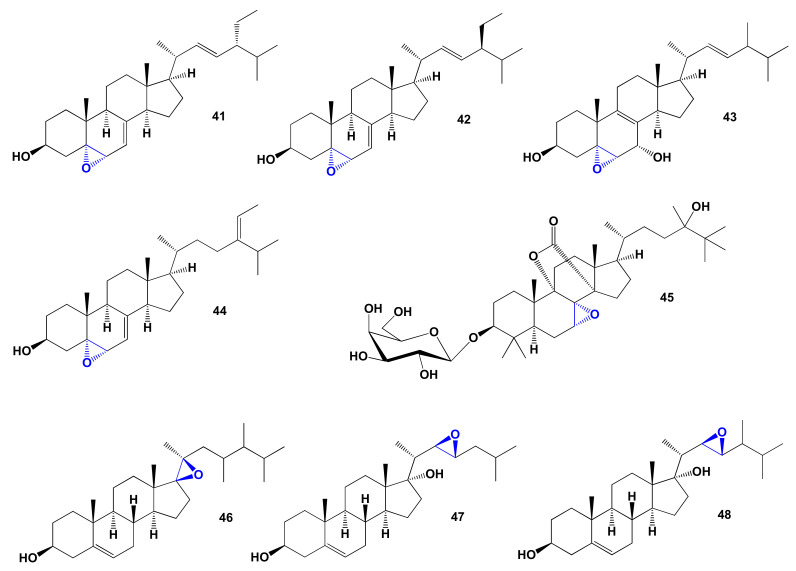
Bioactive α,β-epoxy steroids derived from marine sources.

**Figure 13 marinedrugs-19-00650-f013:**
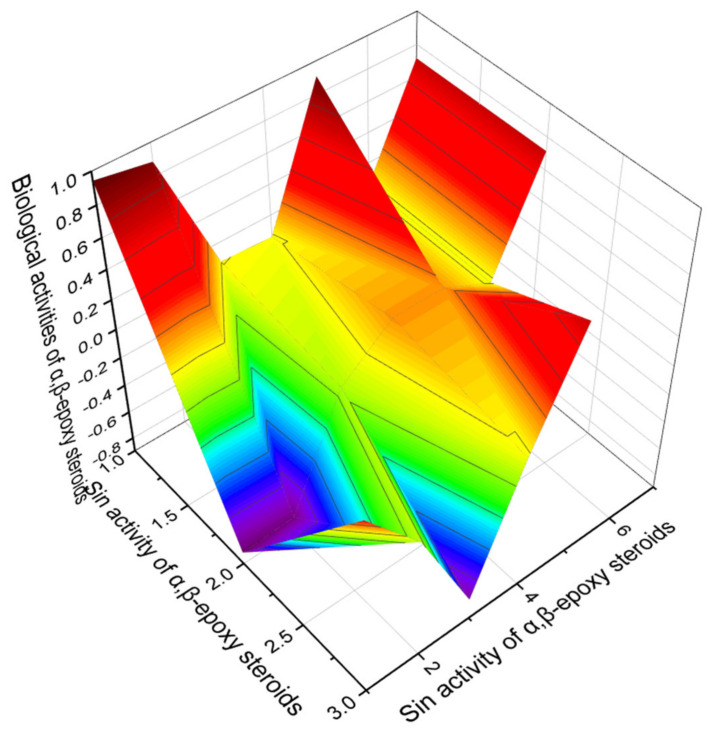
The 3D graph shows the predicted and calculated biological activity of α,β-epoxy steroids (compound numbers: **42**, **44**, and **48**) showing the highest degree of confidence, more than 92.4%.

**Figure 14 marinedrugs-19-00650-f014:**
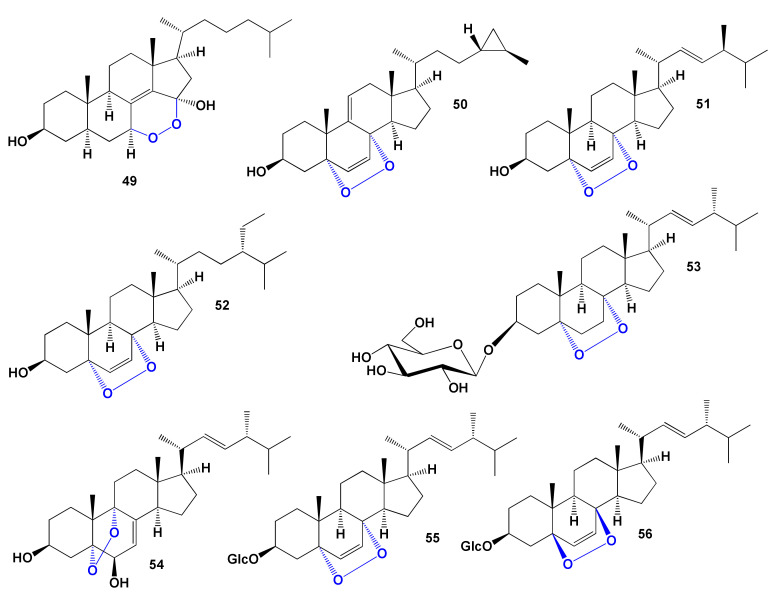
Bioactive steroid endoperoxides derived from marine sources and fungi.

**Figure 15 marinedrugs-19-00650-f015:**
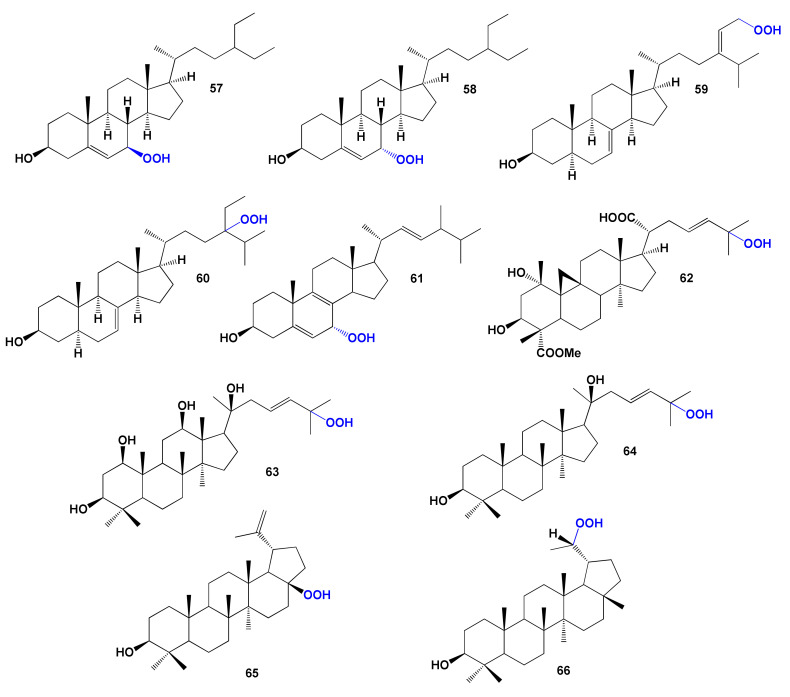
Bioactive steroid hydroperoxides derived from plants.

**Figure 16 marinedrugs-19-00650-f016:**
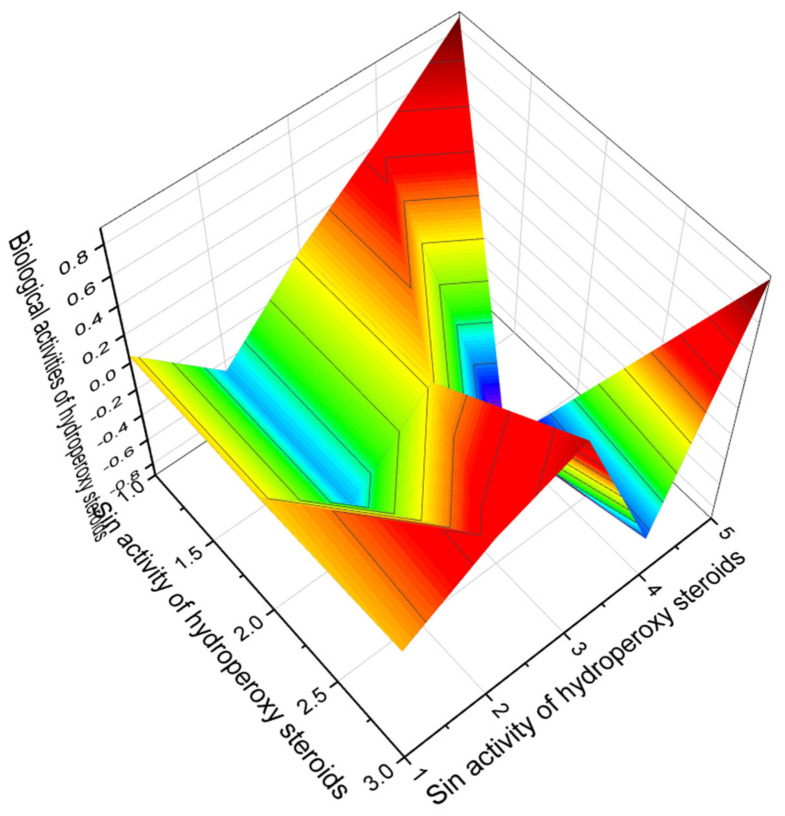
The 3D graph shows the predicted and calculated biological activity of steroid hydroperoxides (compound numbers: **59**, **63**, and **66**) showing the highest degree of confidence, more than 93%.

**Figure 17 marinedrugs-19-00650-f017:**
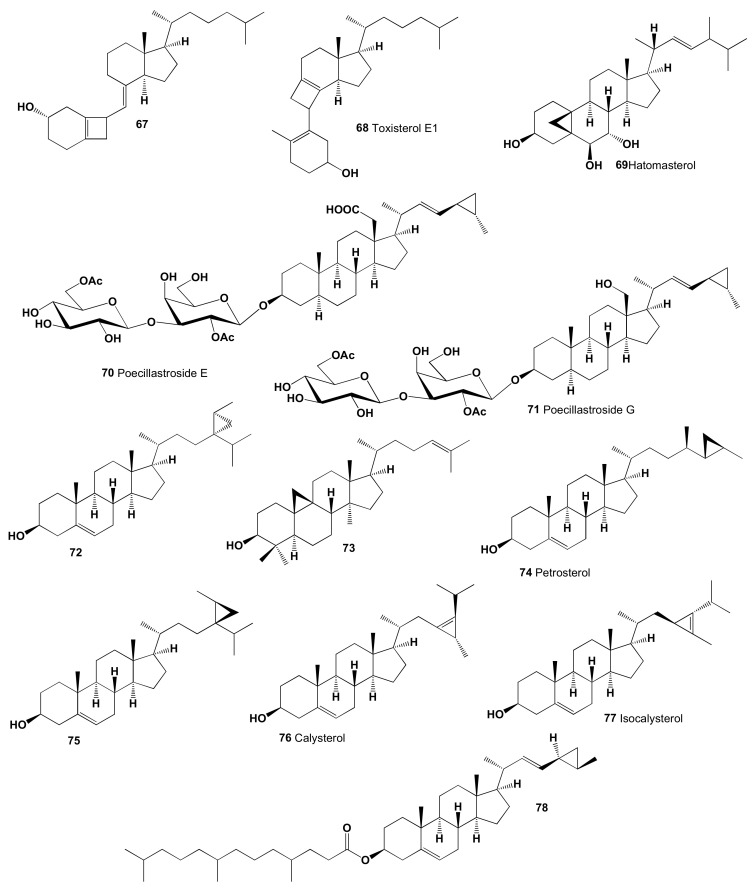
Bioactive CBS derived from fungi and marine sources.

**Figure 18 marinedrugs-19-00650-f018:**
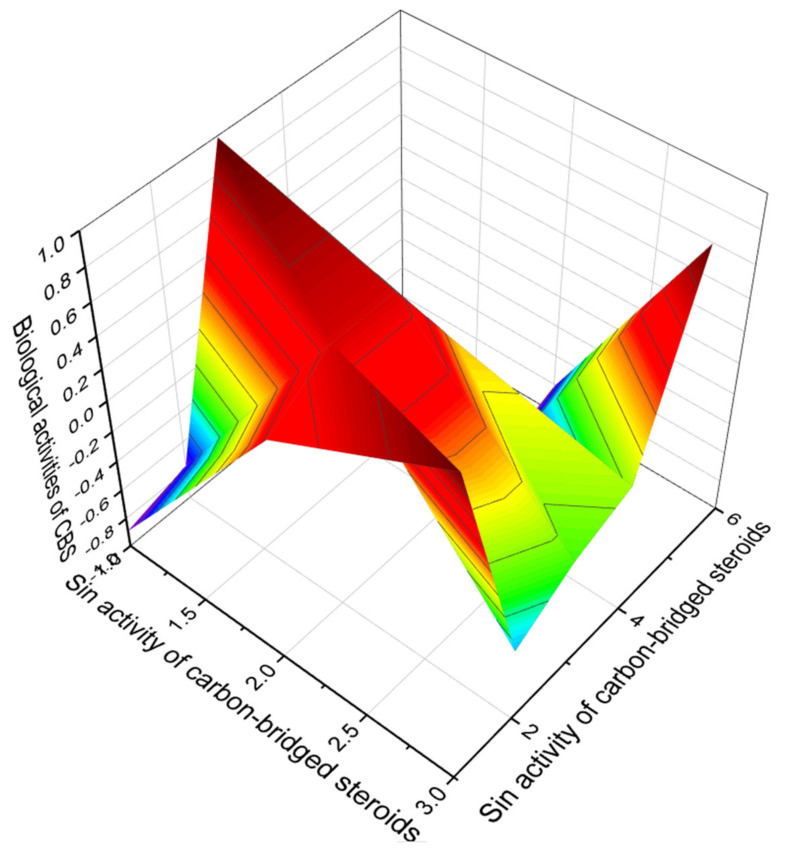
The 3D graph shows the predicted and calculated biological activity of carbon-bridged steroids (compound numbers: **70**, **74**, and **78**) showing the highest degree of confidence, more than 95%.

**Figure 19 marinedrugs-19-00650-f019:**
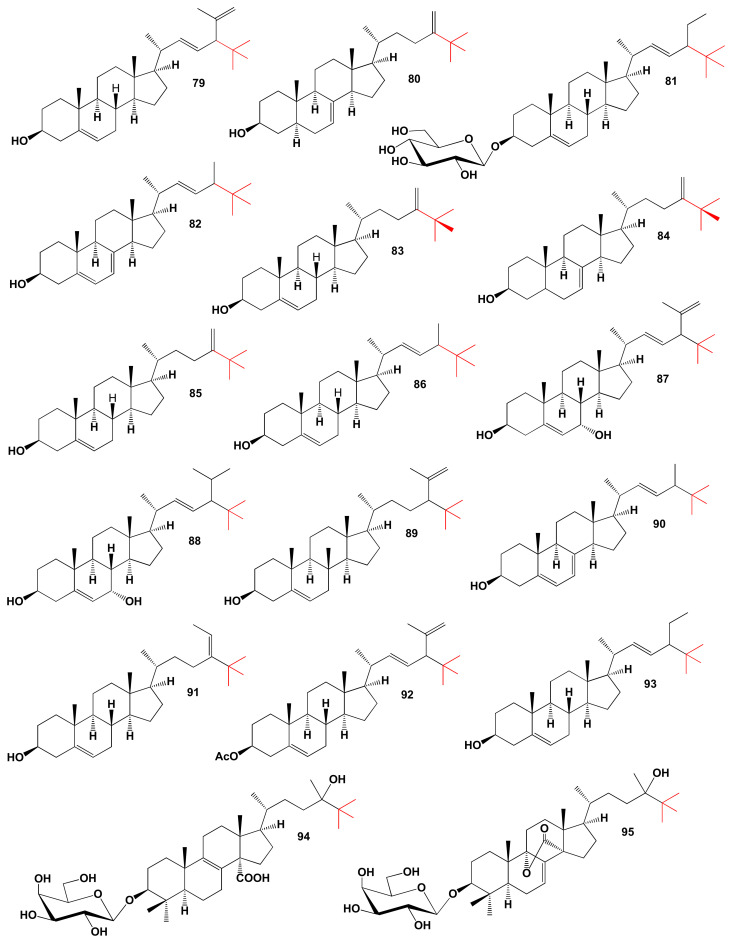
Bioactive neo steroids derived from plants and marine sources.

**Figure 20 marinedrugs-19-00650-f020:**
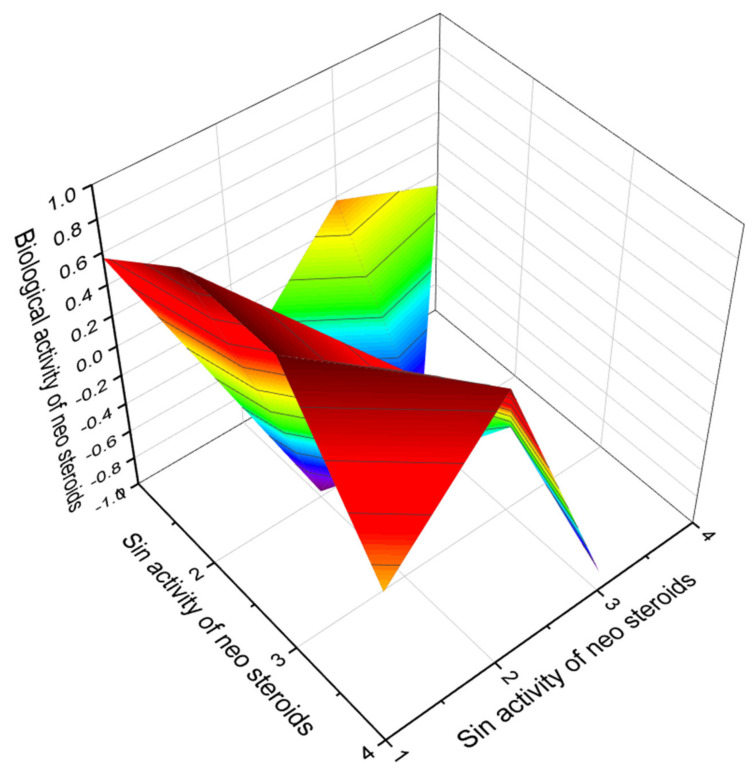
The 3D graph shows the predicted and calculated biological activity of neo steroids (compound numbers: **81**, **86**, **89**, and **93**) showing the highest degree of confidence, more than 96.9%.

**Figure 21 marinedrugs-19-00650-f021:**
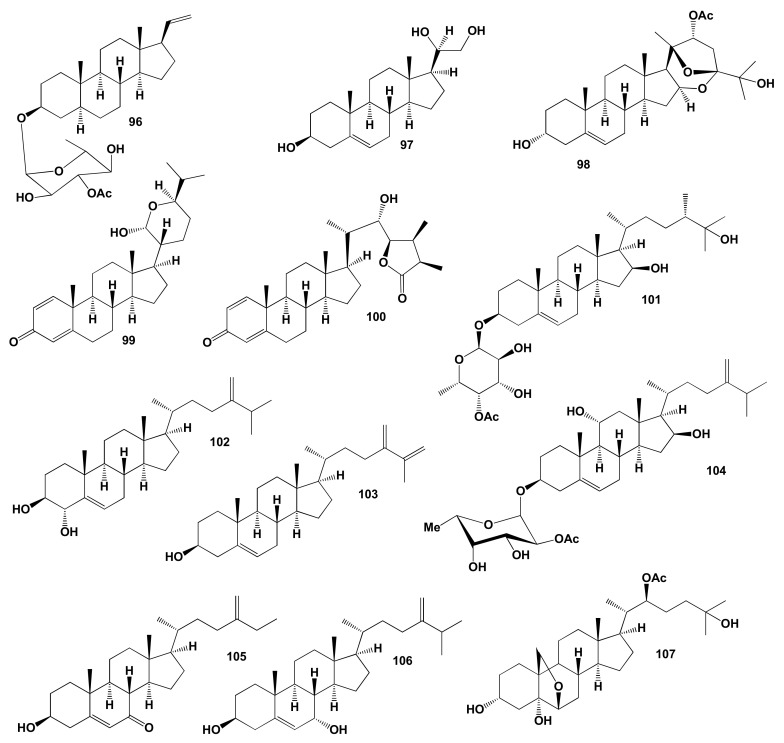
Bioactive steroids derived from soft corals.

**Figure 22 marinedrugs-19-00650-f022:**
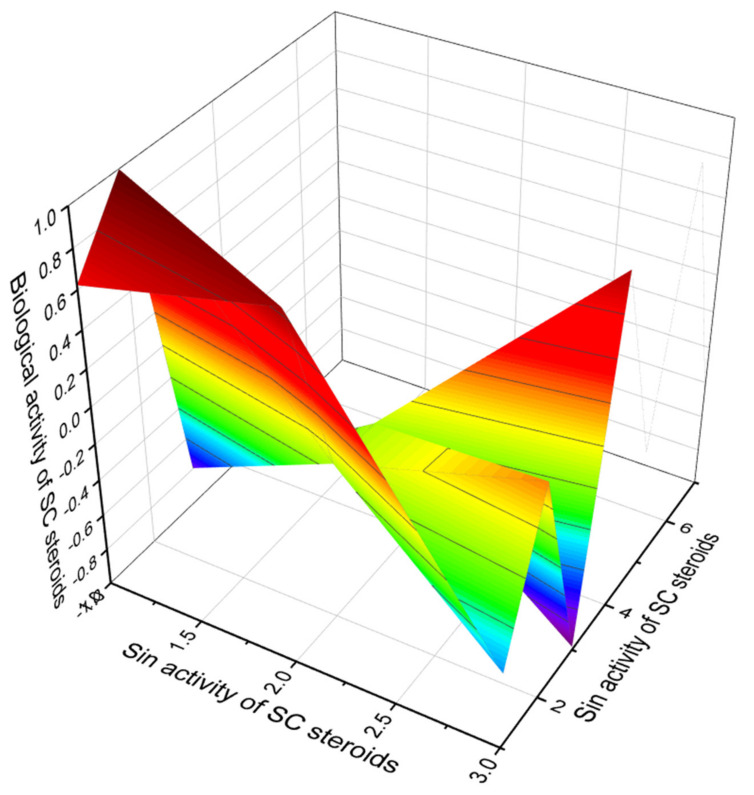
The 3D graph shows the predicted and calculated biological activity of soft coral steroids (compound numbers: **99**, **101** and **104**) showing the highest degree of confidence, more than 96.2%.

**Figure 23 marinedrugs-19-00650-f023:**
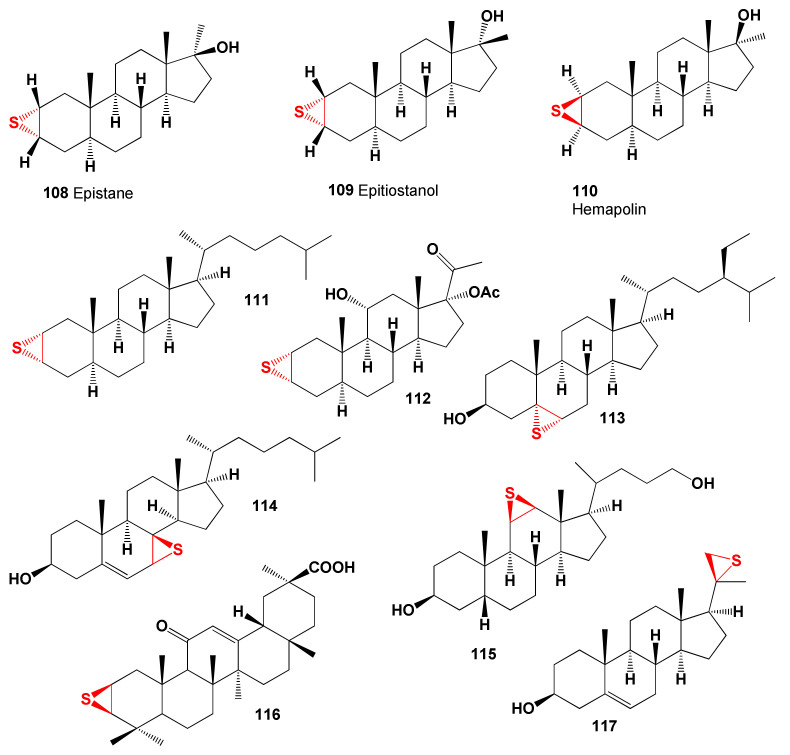
Bioactive epithio steroids and triterpenoid containing the thiirane group in a variety of steroid backbones.

**Figure 24 marinedrugs-19-00650-f024:**
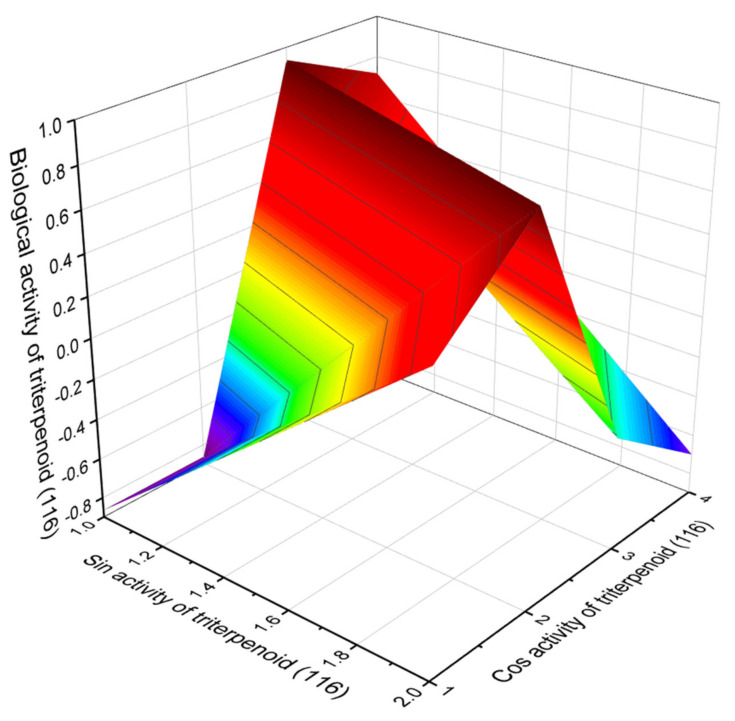
The 3D graph shows the predicted and calculated biological activity of triterpenod (**116**), which was synthesized from a natural sample of 18β-glycyrrhetinic acid, showing the highest degree of confidence as lipid metabolism regulator properties, more than 95.4%. 18β-Glycyrrhetinic acid is a β-amyrin-type pentacyclic triterpenoid derived from the herb licorice (*Glycyrrhiza glabra*). It is known and used as a flavoring agent and masks the bitter taste of drugs such as aloe and quinine; it is also effective in the treatment of peptic ulcer disease and has some additional pharmacological properties with possible antiviral, antifungal, antiprotozoal, and antibacterial effects.

**Figure 25 marinedrugs-19-00650-f025:**
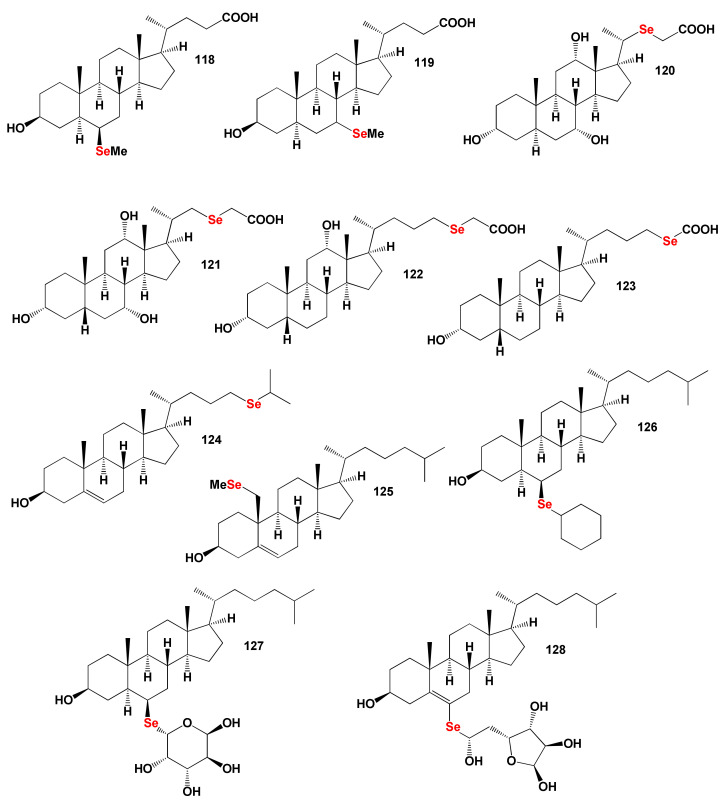
Bioactive synthetic selena steroids.

**Figure 26 marinedrugs-19-00650-f026:**
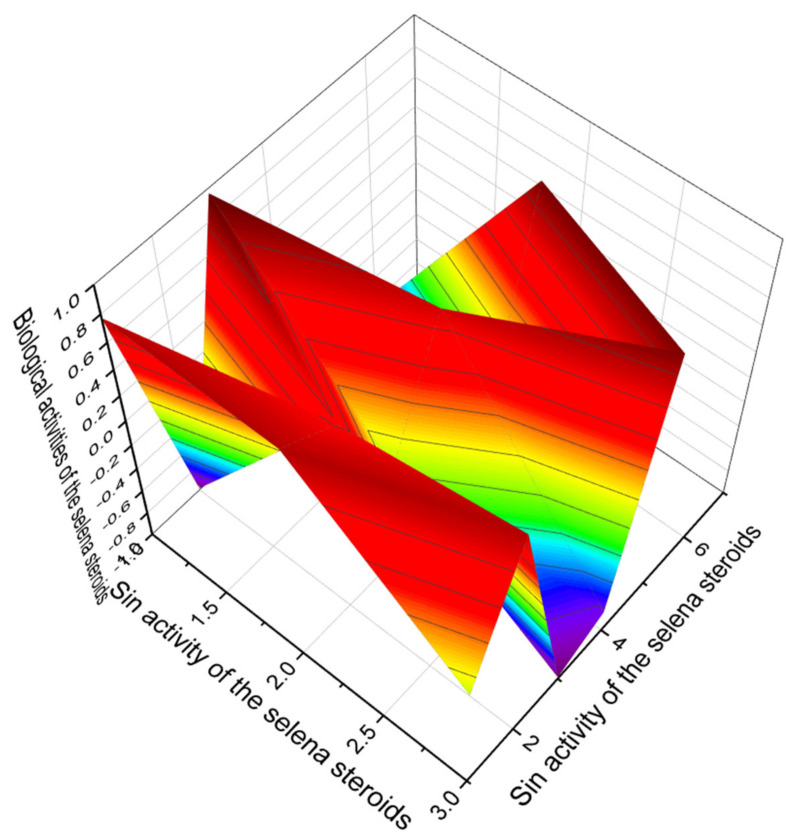
The 3D graph shows the predicted and calculated biological activity of the selena steroids (compound numbers: **120**, **121** and **122**) showing the highest degree of confidence, more than 99.5%.

**Figure 27 marinedrugs-19-00650-f027:**
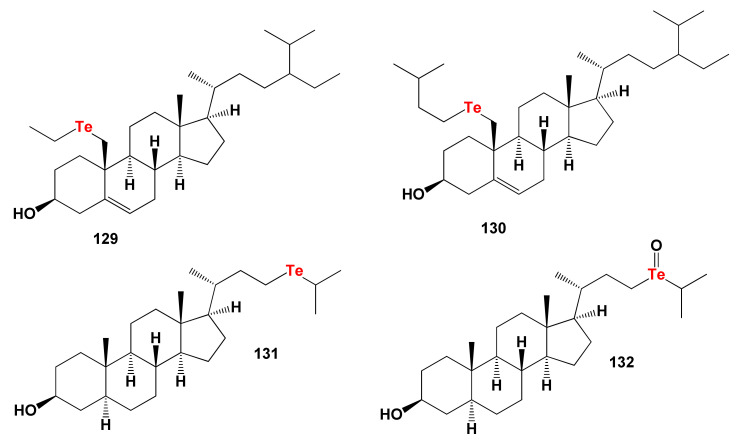
Bioactive synthetic tellura steroids.

**Figure 28 marinedrugs-19-00650-f028:**
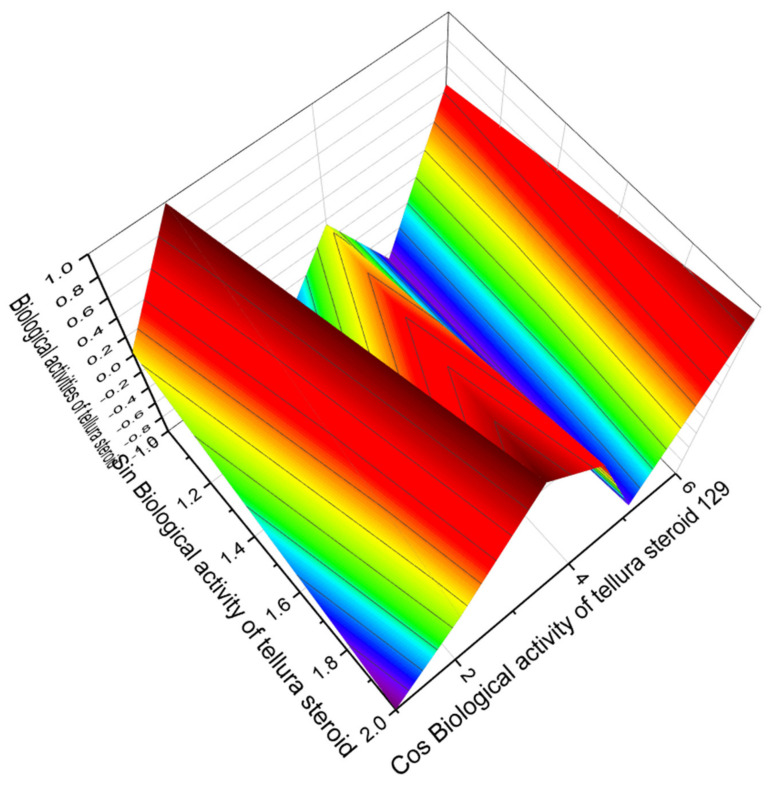
The 3D graph shows the predicted and calculated biological activity of the tellura steroid (**129**) showing the highest degree of confidence, more than 99.5%. This tellura steroid is interesting, in that it is a rare case when any chemical compound shows simultaneously such activities as prevention and treatment of neurodegenerative diseases Alzheimer’s and Parkinson’s with strong confidence, over 94 percent. In addition, this steroid demonstrated antioxidant and antihypercholesterolemic activities, and can also be used as a potential drug for the treatment of atherosclerosis. The maximum values of various biological activities are shown in red.

**Figure 29 marinedrugs-19-00650-f029:**
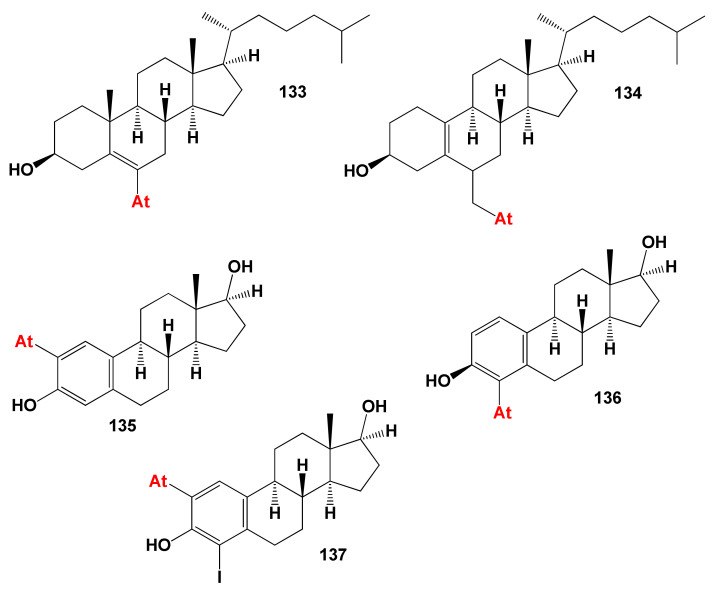
Biological active synthetic astatosteroids.

**Figure 30 marinedrugs-19-00650-f030:**
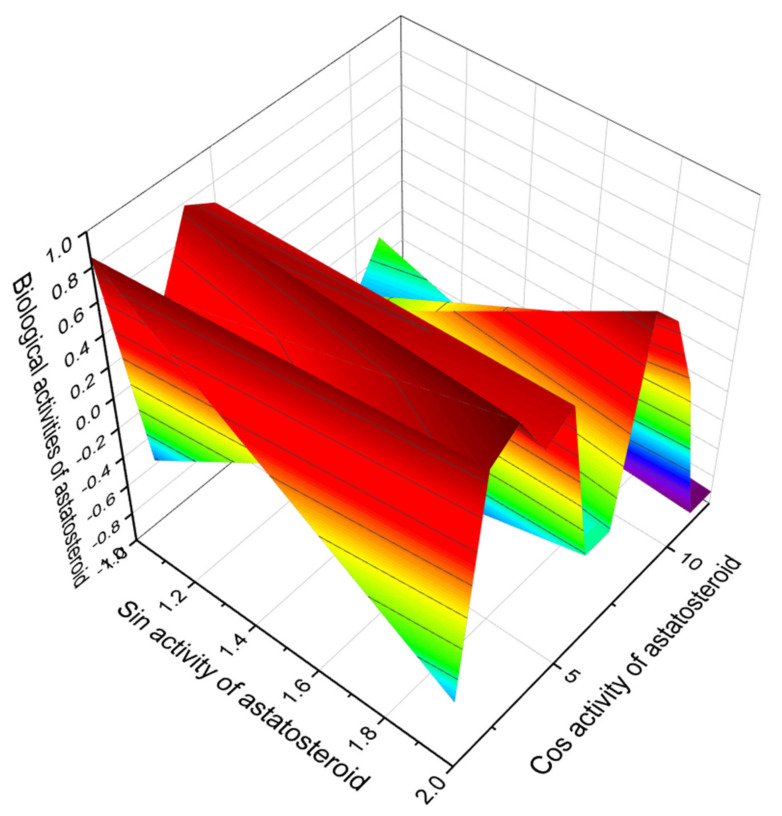
The 3D graph shows the predicted and calculated biological activity of the astatosteroid (**133**) showing the highest degree of confidence, more than 97.6%. Moreover, it can also be used as a potential drug for the treatment of bone diseases.

**Table 1 marinedrugs-19-00650-t001:** Biological activities of aromatic steroids with over 90% confidence.

No.	Discovered Activity, (Pa) *	Reported Activity	References
**1**	Ovulation inhibitor (0.942) Cardiovascular analeptic (0.924) Antihypercholesterolemic (0.871) Lipid metabolism regulator (0.730)	Inhibitor aromatase Sulfatase inhibitor Estrogenic Promotor breast cancer	[81,82,83,84]
**2**	Antihypercholesterolemic (0.904) Ovulation inhibitor (0.889) Neuroprotector (0.870) Anesthetic general (0.868) Acute neurologic disorders treatment (0.793) Prostate disorders treatment (0.729) Anti-inflammatory (0.713)	Antioxidant Anti-inflammatory Uterotrophic RNA polymerase Promoter of breast, ovarian and endometrial cancers Neuroprotective properties	[81,85,86,87]
**3**	Ovulation inhibitor (0.900) Acute neurologic disorders treatment (0.822) Antihypercholesterolemic (0.791)	Estrogenic Agonist of the ERs RNA polymerase	[88,89]
**4**	Antihypercholesterolemic (0.856) Ovulation inhibitor (0.847) Cardiovascular analeptic (0.842) Lipid metabolism regulator (0.788)	Estrogenic Estrogen agonist	[90,91,92]
**5**	Acute neurologic disorders treatment (0.912) Male reproductive disfunction treatment (0.847) Ovulation inhibitor (0.786) Postmenopausal disorders treatment (0.643) Antihypercholesterolemic (0.640) Menopausal disorders treatment (0.579)	Inhibitor aromatase Sulfatase inhibitor Inhibitor of human breast cancer Concentration Cardiovascular agent Postmenopausal disorders	[93,94]
**6**	Antihypercholesterolemic (0.973) Acute neurologic disorders treatment (0.922) Lipid metabolism regulator (0.907) Antithrombotic (0.714) Ovulation inhibitor (0.662) Dementia treatment (0.616) Hypolipemic (0.613) Male reproductive disfunction treatment (0.587) Menopausal disorders treatment (0.582)	Estrogenic Ovarian activity Urine production	[95]
**7**	Ovulation inhibitor (0.956) Cardiovascular analeptic (0.927) Antihypercholesterolemic (0.868) Male reproductive disfunction treatment (0.847) Menopausal disorders treatment (0.842) Acute neurologic disorders treatment (0.745) Lipid metabolism regulator (0.701) Menstruation disorders treatment (0.639) Postmenopausal disorders treatment (0.605) Muscular dystrophy treatment (0.601) Contraceptive female (0.570) Anti-infertility, female (0.567)	Estrogenic Antiestrogenic effects Anticancer	[96,97,98]
**8**	Ovulation inhibitor (0.930) Cardiovascular analeptic (0.925) Antihypercholesterolemic (0.855) Male reproductive disfunction treatment (0.843) Acute neurologic disorders treatment (0.780) Menopausal disorders treatment (0.747) Lipid metabolism regulator (0.601) Muscular dystrophy treatment (0.579) Postmenopausal disorders treatment (0.561) Menstruation disorders treatment (0.533)	Estrogenic Strongest neuroprotective effect UDP-glucuronosyltransferase Anti-breast cancer	[99,100,101]
**9**	Ovulation inhibitor (0.953) Cardiovascular analeptic (0.928) Antihypercholesterolemic (0.857) Menopausal disorders treatment (0.807) Male reproductive disfunction treatment (0.805) Acute neurologic disorders treatment (0.660) Contraceptive (0.655) Lipid metabolism regulator (0.645) Contraceptive female (0.558) Menstruation disorders treatment (0.542) Anti-infertility, female (0.503) Postmenopausal disorders treatment (0.500)	Antioxidant Estrogenic Anti-breast cancer	[102]
**10**	Ovulation inhibitor (0.925) Cardiovascular analeptic (0.922) Antihypercholesterolemic (0.833) Male reproductive disfunction treatment (0.821) Menopausal disorders treatment (0.712) Acute neurologic disorders treatment (0.662) Contraceptive (0.602) Lipid metabolism regulator (0.571) Muscular dystrophy treatment (0.548)	Estrogenic Proliferation of human breast cancer	[103,104,105]
**11**	Antihypercholesterolemic (0.959) Hypolipemic (0.808)	Activity not studied	
**12**	Lipid metabolism regulator (0.913) Antihypercholesterolemic (0.767)	Activity not studied	
**13**	Antihypercholesterolemic (0.961) Hypolipemic (0.711)	Activity not studied	
**14**	Antihypercholesterolemic (0.953)	Activity not studied	
**15**	Antihypercholesterolemic (0.946)	Activity not studied	
**16**	Antihypercholesterolemic (0.907)	Activity not studied	
**17**	Antihypercholesterolemic (0.929)	Activity not studied	
**18**	Antihypercholesterolemic (0.935)	Activity not studied	
**19**	Antihypercholesterolemic (0.950)	Activity not studied	
**20**	Antihypercholesterolemic (0.914)	Anticancer	[52]

* Only activities with Pa > 0.5 are shown.

**Table 2 marinedrugs-19-00650-t002:** Predicted biological activities of steroid phosphate esters.

No.	Lipid Metabolism Regulators, (Pa) *	Reported Activity	Ref.
**21**	Wound healing agent (0.975) Hepatoprotectant (0.961) Analeptic (0.952) Antihypercholesterolemic (0.926) Cholesterol synthesis inhibitor (0.799)	Activity not studied	
**22**	Wound healing agent (0.947) Analeptic (0.941) Hepatoprotectant (0.932) Anticarcinogenic (0.915) Antihypercholesterolemic (0.912) Cholesterol synthesis inhibitor (0.778)	Activity not studied	
**23**	Antihypercholesterolemic (0.900) Hepatoprotectant (0.853) Wound healing agent (0.844) Antineoplastic (0.816) Antiinflammatory (0.782) Cholesterol synthesis inhibitor (0.778) Atherosclerosis treatment (0.675)	Activity not studied	
**24**	Neuroprotector (0.987) Anesthetic general (0.959) Respiratory analeptic (0.944) Antihypercholesterolemic (0.909)	Substrate for phosphatases	[112]
**25**	Anesthetic general (0.991) Respiratory analeptic (0.990) Neuroprotector (0.976) Antiinflammatory (0.906) Antihypercholesterolemic (0.900)	Increases blood sugar levels	[113]
**26**	Respiratory analeptic (0.979) Anesthetic general (0.973) Neuroprotector (0.972) Antihypercholesterolemic (0.971) Wound healing agent (0.913) Antineoplastic (0.826) Cholesterol synthesis inhibitor (0.801)	Stabilizes blood pressure	[114]
**27**	Respiratory analeptic (0.995) Anesthetic general (0.948) Antihypercholesterolemic (0.945) Neuroprotector (0.932) Hemostatic (0.910) Wound healing agent (0.897) Cholesterol synthesis inhibitor (0.867) Acute neurologic disorders treatment (0.827)	Reduces cholesterol levels	[116]
**28**	Antihypercholesterolemic (0.967) Wound healing agent (0.921) Neuroprotector (0.909) Cholesterol synthesis inhibitor (0.872)	Reduces cholesterol levels	[116]
**29**	Antihypercholesterolemic (0.996) Cholesterol absorption inhibitor (0.976) Cholesterol synthesis inhibitor (0.952) Lipid metabolism regulator (0.952) Lipoprotein disorders treatment (0.893)	Cholesterol biosynthesis inhibitor	[117]
**30**	Antihypercholesterolemic (0.999) Antihyperlipoproteinemic (0.986) Hypolipemic (0.974) Cholesterol absorption inhibitor (0.957) Lipid metabolism regulator (0.954) Cholesterol synthesis inhibitor (0.916) Lipoprotein disorders treatment (0.782) Acute neurologic disorders treatment (0.751) Atherosclerosis treatment (0.729)	Cholesterol biosynthesis inhibitor	[117]
**31**	Neuroprotector (0.982) Anesthetic general (0.931) Antihypercholesterolemic (0.909) Acute neurologic disorders treatment (0.831) Prostate disorders treatment (0.640)	Remedy for the treatment of prostate cancer	[118]

* Only activities with Pa > 0.5 are shown.

**Table 3 marinedrugs-19-00650-t003:** Predicted biological activities of secosteroids.

No.	Discovered Activity, (Pa) *	Reported Activity	Ref.
**32**	Antihypercholesterolemic (0.912) Hypolipemic (0.802) Atherosclerosis treatment (0.643) Antiparkinsonian, rigidity relieving (0.562)	Activity not studied	
**33**	Antihypercholesterolemic (0.905) Hypolipemic (0.753) Atherosclerosis treatment (0.559)	Activity not studied	
**34**	Antihypercholesterolemic (0.908) Antineoplastic (0.785) Hypolipemic (0.764) Apoptosis agonist (0.747) Cholesterol synthesis inhibitor (0.744)	Anticancer	[131]
**35**	Antihypercholesterolemic (0.916) Antineoplastic (0.833) Hypolipemic (0.827) Apoptosis agonist (0.771) Cholesterol synthesis inhibitor (0.685) Atherosclerosis treatment (0.633)	Anticancer	[131]
**36**	Antihypercholesterolemic (0.904) Hypolipemic (0.767) Antineoplastic (0.743) Apoptosis agonist (0.676) Proliferative diseases treatment (0.625) Atherosclerosis treatment (0.539)	Anticancer	[133]
**37**	Antihypercholesterolemic (0.915) Lipid metabolism regulator (0.768)	Activity not studied	
**38**	Antihypercholesterolemic (0.909) Hypolipemic (0.786)	Anticancer	[147]
**39**	Antiparkinsonian, rigidity relieving (0.960) Hyperparathyroidism treatment (0.892) Antihypercholesterolemic (0.845) Hypolipemic (0.790) Atherosclerosis treatment (0.628)	Calcium and phosphates metabolism regulator	[148]
**40**	Chemopreventive (0.989) Hepatoprotectant (0.986) Respiratory analeptic (0.978) Antihypercholesterolemic (0.977) Proliferative diseases treatment (0.969) Antimycobacterial (0.939) Neuroprotector (0.895) Antineoplastic (0.874) Antiprotozoal (Leishmania) (0.772) Atherosclerosis treatment (0.601) Neurodegenerative diseases treatment (0.590) Alzheimer’s disease treatment (0.570)	Activity not studied previously	

* Only activities with Pa > 0.5 are shown.

**Table 4 marinedrugs-19-00650-t004:** Predicted biological activities of α,β-epoxy steroids.

No.	Discovered Activity, (Pa) *	Reported Activity	Ref.
**41**	Apoptosis agonist (0.950) Antihypercholesterolemic (0.931) Antineoplastic (0.886) Antieczematic (0.842) Atherosclerosis treatment (0.712)	Cytotoxic	[160]
**42**	Apoptosis agonist (0.950) Antihypercholesterolemic (0.931) Antineoplastic (0.886) Antieczematic (0.842) Atherosclerosis treatment (0.712)	Cytotoxic	[160]
**43**	Apoptosis agonist (0.954) Antineoplastic (0.914) Antihypercholesterolemic (0.906) Atherosclerosis treatment (0.741)	Activity not studied	
**44**	Antihypercholesterolemic (0.934) Apoptosis agonist (0.929) Hypolipemic (0.864) Antineoplastic (0.861)	Activity not studied	
**45**	Hepatoprotectant (0.994) Respiratory analeptic (0.990) Antihypercholesterolemic (0.897)	Activity not studied	
**46**	Antihypercholesterolemic (0.900) Neuroprotector (0.749) Cholesterol synthesis inhibitor (0.636)	Activity not studied	
**47**	Antihypercholesterolemic (0.901) Lipid metabolism regulator (0.833) Prostate disorders treatment (0.714)	Anticancer	[165]
**48**	Antihypercholesterolemic (0.924) Lipid metabolism regulator (0.820) Neuroprotector (0.728)	Anticancer	[165]

* Only activities with Pa > 0.5 are shown.

**Table 5 marinedrugs-19-00650-t005:** Biological activities of endoperoxy steroids.

No.	Lipid Metabolism Regulators, (Pa) *	Reported Activity	Ref.
**49**	Antihypercholesterolemic (0.914) Hypolipemic (0.635)	Activity not studied	
**50**	Atherosclerosis treatment (0.911) Hypolipemic (0.836) Lipoprotein disorders treatment (0.826) Antihypercholesterolemic (0.802)	Activity not studied	
**51**	Atherosclerosis treatment (0.907) Antihypercholesterolemic (0.788)	Activity not studied	
**52**	Atherosclerosis treatment (0.919) Hypolipemic (0.822) Lipoprotein disorders treatment (0.814)	Activity not studied	
**53**	Antihypercholesterolemic (0.926) Hypolipemic (0.800) Atherosclerosis treatment (0.709)	Activity not studied	
**54**	Antihypercholesterolemic (0.900) Hypolipemic (0.827) Atherosclerosis treatment (0.659) Hyperparathyroidism treatment (0.502)	Activity not studied	
**55**	Antihypercholesterolemic (0.917) Hypolipemic (0.786)	Anticancer	[183]
**56**	Antihypercholesterolemic (0.917) Atherosclerosis treatment (0.858) Hypolipemic (0.786)	Anticancer	[183]

* Only activities with Pa > 0.5 are shown.

**Table 6 marinedrugs-19-00650-t006:** Predicted biological activities of hydroperoxy steroids.

No.	Discovered Activity, (Pa) *	Reported Activity	Ref.
**57**	Antihypercholesterolemic (0.905) Cholesterol synthesis inhibitor (0.799)	Activity not studied	
**58**	Antihypercholesterolemic (0.905) Hypolipemic (0.802) Cholesterol synthesis inhibitor (0.621) Atherosclerosis treatment (0.542)	Activity not studied	
**59**	Antihypercholesterolemic (0.933) Hypolipemic (0.877) Cholesterol synthesis inhibitor (0.644) Atherosclerosis treatment (0.675) Prostate disorders treatment (0.645)	Cytotoxic	[186]
**60**	Antihypercholesterolemic (0.933) Hypolipemic (0.877) Antineoplastic (0.835) Cholesterol synthesis inhibitor (0.650) Atherosclerosis treatment (0.554)	Cytotoxic	[186]
**61**	Antihypercholesterolemic (0.922) Hypolipemic (0.868) Atherosclerosis treatment (0.678) Antiparkinsonian, rigidity relieving (0.516)	Activity not studied	
**62**	Antihypercholesterolemic (0.913) Antineoplastic (0.802) Hypolipemic (0.795)	Hepatoprotective Cytotoxic	[191]
**63**	Antihypercholesterolemic (0.933) Hypolipemic (0.877) Cholesterol synthesis inhibitor (0.644)	Activity not studied	
**64**	Antihypercholesterolemic (0.923) Hypolipemic (0.774) Cholesterol synthesis inhibitor (0.604) Biliary tract disorders treatment (0.577)	Activity not studied	
**65**	Antihypercholesterolemic (0.918) Hypolipemic (0.779) Biliary tract disorders treatment (0.655)	Antiproliferative	[194]
**66**	Antihypercholesterolemic (0.930) Hypolipemic (0.767) Biliary tract disorders treatment (0.717) Atherosclerosis treatment (0.590)	Activity not studied	

* Only activities with Pa > 0.5 are shown.

**Table 7 marinedrugs-19-00650-t007:** Predicted biological activities of CBS steroids.

No.	Discovered Activity, (Pa) *	Reported Activity	Ref.
**67**	Antihypercholesterolemic (0.902) Hypolipemic (0.721) Cholesterol synthesis inhibitor (0.534)	Activity not studied	
**68**	Antihypercholesterolemic (0.932) Hypolipemic (0.695) Cholesterol synthesis inhibitor (0.588)	Activity not studied	
**69**	Antineoplastic (0.915) Antihypercholesterolemic (0.900) Hypolipemic (0.897) Apoptosis agonist (0.892) Antineoplastic (liver cancer) (0.822) Chemopreventive (0.776) Atherosclerosis treatment (0.690) Cytoprotectant (0.611) Prostate cancer treatment (0.557) Antimetastatic (0.528)	Cytotoxic	[206]
**70**	Antihypercholesterolemic (0.953) Hypolipemic (0.758) Lipid metabolism regulator (0.674) Atherosclerosis treatment (0.513)	Antifungal	[207]
**71**	Antihypercholesterolemic (0.939) Hypolipemic (0.746) Lipid metabolism regulator (0.599)	No activity detected	[207]
**72**	Antihypercholesterolemic (0.923) Hypolipemic (0.732) Atherosclerosis treatment (0.643) Cholesterol synthesis inhibitor (0.640)	Activity not studied	
**73**	Hypolipemic (0.900) Atherosclerosis treatment (0.689) Cholesterol synthesis inhibitor (0.671) Antihypercholesterolemic (0.662) Lipid metabolism regulator (0.529)	Activity not studied	
**74**	Antihypercholesterolemic (0.964) Hypolipemic (0.849) Antineoplastic (0.849) Antihyperlipoproteinemic (0.801) Cholesterol synthesis inhibitor (0.671) Atherosclerosis treatment (0.610) Prostate cancer treatment (0.601)	Cytotoxic Anticancer	[211,212,213,214]
**75**	Antihypercholesterolemic (0.923) Hypolipemic (0.732) Atherosclerosis treatment (0.643) Cholesterol synthesis inhibitor (0.640)	Activity not studied	
**76**	Antihypercholesterolemic (0.935) Hypolipemic (0.731) Antihyperlipoproteinemic (0.689) Cholesterol synthesis inhibitor (0.600)	Activity not studied	
**77**	Antihypercholesterolemic (0.908) Hypolipemic (0.726) Cholesterol synthesis inhibitor (0.589) Antihyperlipoproteinemic (0.587)	Activity not studied	
**78**	Antihypercholesterolemic (0.969) Hypolipemic (0.810) Lipid metabolism regulator (0.716) Cholesterol synthesis inhibitor (0.707) Atherosclerosis treatment (0.586)	Activity not studied	

* Only activities with Pa > 0.5 are shown.

**Table 8 marinedrugs-19-00650-t008:** Biological activities of neo steroids.

No.	Discovered Activity, (Pa) *	Reported Activity	Ref.
**79**	Antihypercholesterolemic (0.964) Atherosclerosis treatment (0.717)	Activity not studied	
**80**	Antihypercholesterolemic (0.956)	Activity not studied	
**81**	Antihypercholesterolemic (0.989) Hypolipemic (0.808)	Activity not studied	
**82**	Antihypercholesterolemic (0.960) Atherosclerosis treatment (0.683)	Activity not studied	
**83**	Antihypercholesterolemic (0.961) Cholesterol synthesis inhibitor (0.745)	Activity not studied	
**84**	Antihypercholesterolemic (0.956) Cholesterol synthesis inhibitor (0.747)	Activity not studied	
**85**	Antihypercholesterolemic (0.964)	Activity not studied	
**86**	Antihypercholesterolemic (0.969)	Activity not studied	
**87**	Antihypercholesterolemic (0.952) Atherosclerosis treatment (0.710)	Activity not studied	
**88**	Antihypercholesterolemic (0.965) Atherosclerosis treatment (0.677)	Activity not studied	
**89**	Antihypercholesterolemic (0.969)	Activity not studied	
**90**	Antihypercholesterolemic (0.965) Atherosclerosis treatment (0.700)	Activity not studied	
**91**	Antihypercholesterolemic (0.969)	Activity not studied	
**92**	Antihypercholesterolemic (0.962) Atherosclerosis treatment (0.704) Lipid metabolism regulator (0.702)	Activity not studied	
**93**	Antihypercholesterolemic (0.974) Antimicrobial treatment (0.717)	Antimicrobial	[237]
**94**	Antihypercholesterolemic (0.961) Antineoplastic (0.863) Anticarcinogenic (0.828) Lipid metabolism regulator (0.747) Antimetastatic (0.590)	Anticancer	[238,239]
**95**	Antihypercholesterolemic (0.936) Antineoplastic (0.870) Anticarcinogenic (0.807) Antimetastatic (0.598)	Anticancer	[238,239]

* Only activities with Pa > 0.5 are shown.

**Table 9 marinedrugs-19-00650-t009:** Biological activities of soft coral steroids.

No.	Discovered Activity, (Pa) *	Reported Activity	Ref.
**96**	Neuroprotector (0.983) Antihypercholesterolemic (0.919) Acute neurologic disorders treatment (0.636) Hypolipemic (0.626)	Anticancer	[243]
**97**	Antihypercholesterolemic (0.954) Neuroprotector (0.754) Hypolipemic (0.704)	Activity not studied	
**98**	Antihypercholesterolemic (0.927) Neuroprotector (0.773) Hypolipemic (0.584)	Activity not studied	
**99**	Antihypercholesterolemic (0.962)	Activity not studied	
**100**	Antihypercholesterolemic (0.918) Anti-inflammatory (0.903) Antineoplastic (0.875) Proliferative diseases treatment (0.856) Atherosclerosis treatment (0.618)	Anti-inflammatory Anticancer	[247]
**101**	Antihypercholesterolemic (0.976) Hypolipemic (0.735) Lipid metabolism regulator (0.707)	Activity not studied	
**102**	Antihypercholesterolemic (0.920) Hypolipemic (0.798) Atherosclerosis treatment (0.637) Hyperparathyroidism treatment (0.523)	Activity not studied	
**103**	Antihypercholesterolemic (0.955) Atherosclerosis treatment (0.596)	Activity not studied	
**104**	Proliferative diseases treatment (0.967) Chemopreventive (0.958) Antihypercholesterolemic (0.947) Anticarcinogenic (0.907) Antineoplastic (0.902) Hypolipemic (0.781) Atherosclerosis treatment (0.609)	Anticancer	[251]
**105**	Antihypercholesterolemic (0.957) Hypolipemic (0.809) Atherosclerosis treatment (0.592)	Activity not studied	
**106**	Antihypercholesterolemic (0.955) Hypolipemic (0.878) Atherosclerosis treatment (0.658)	Activity not studied	
**107**	Neuroprotector (0.938) Antihypercholesterolemic (0.912) Atherosclerosis treatment (0.548)	Activity not studied	

* Only activities with Pa > 0.5 are shown.

**Table 10 marinedrugs-19-00650-t010:** Biological activities of epithio steroids.

No.	Discovered Activity, (Pa) *	Reported Activity	Ref.
**108**	Antineoplastic (0.964) Antisecretoric (0.948) Estrogen antagonist (0.860) Cardiotonic (0.729) Prostate disorders treatment (0.709) Neuroprotector (0.723) Bone diseases treatment (0.693) Antineoplastic (breast cancer) (0.598)	Anti-breast cancer Estrogen receptor antagonist	[294]
**109**	Antineoplastic (0.966) Antisecretoric (0.952) Estrogen antagonist (0.832) Anti-inflammatory (0.754) Prostate disorders treatment (0.736) Prostatic (benign) hyperplasia treatment (0.673)	Estrogen antagonist	[294]
**110**	Antineoplastic (0.966) Antisecretoric (0.952) Estrogen antagonist (0.832) Anti-inflammatory (0.754) Prostate disorders treatment (0.736) Prostatic (benign) hyperplasia treatment (0.673)	Estrogen antagonist	[294]
**111**	Antineoplastic (0.932) Antihypercholesterolemic (0.759) Bone diseases treatment (0.729) Hypolipemic (0.676) Estrogen antagonist (0.660)	Anabolic	[294]
**112**	Antisecretoric (0.967) Estrogen antagonist (0.946) Antineoplastic (0.939) Anabolic (0.823)	Anabolic	[284,285,286]
**113**	Cholesterol antagonist (0.933) Antihypercholesterolemic (0.929) Hypolipemic (0.818) Estrogen antagonist (0.443)	Antiseptic, Germicidal, Fungicidal	[294]
**114**	Cholesterol antagonist (0.946) Antihypercholesterolemic (0.930) Hypolipemic (0.781) Estrogen antagonist (0.465)	Antiseptic, Germicidal, Fungicidal	[294]
**115**	Cholesterol antagonist (0.932) Antihypercholesterolemic (0.900) Cardiotonic (0.886) Choleretic (0.871) Atherosclerosis treatment (0.838) Antineoplastic (0.775) Hypolipemic (0.746) Estrogen antagonist (0.611)	DOCA inhibitor	[294]
**116**	Lipid metabolism regulator (0.954) Antineoplastic (0.924) Apoptosis agonist (0.869) Estrogen antagonist (0.505)	Anticancer	[292,295]
**117**	Cholesterol antagonist (0.916) Antihypercholesterolemic (0.836) Hypolipemic (0.801)	Activity not studied	

* Only activities with Pa > 0.5 are shown.

**Table 11 marinedrugs-19-00650-t011:** Biological activities of the selena steroids.

No.	Discovered Activity, (Pa) *	Reported Activity	Ref.
**118**	Antihypercholesterolemic (0.905)	Activity not studied	
**119**	Choleretic (0.909) Antihypercholesterolemic (0.905)	Activity not studied	
**120**	Hypolipemic (0.995) Atherosclerosis treatment (0.991) Lipoprotein disorders treatment (0.982) Antioxidant (0.973) Erythropoiesis stimulant (0.823) Biliary tract disorders treatment (0.808) Laxative (0.709)	Activity not studied	
**121**	Hypolipemic (0.995) Atherosclerosis treatment (0.989) Lipoprotein disorders treatment (0.980) Antioxidant (0.970) Biliary tract disorders treatment (0.808) Erythropoiesis stimulant (0.730) Laxative (0.683)	Activity not studied	
**122**	Hypolipemic (0.996) Atherosclerosis treatment (0.995) Lipoprotein disorders treatment (0.991) Antioxidant (0.978) Erythropoiesis stimulant (0.756)	Activity not studied	
**123**	Hypolipemic (0.913) Antihypercholesterolemic (0.884) Atherosclerosis treatment (0.822)	Activity not studied	
**124**	Antihypercholesterolemic (0.902) Hypolipemic (0.899) Atherosclerosis treatment (0.791)	Activity not studied	
**125**	Antihypercholesterolemic (0.905)	Activity not studied	
**126**	Antihypercholesterolemic (0.908) Bone diseases treatment (0.772) Hypolipemic (0.769)	Activity not studied	
**127**	Antihypercholesterolemic (0.912) Antineoplastic (0.884) Anticarcinogenic (0.753)	Cytotoxic activity Agent for Alzheimer’s disease	[303,310]
**128**	Antihypercholesterolemic (0.953) Antineoplastic (0.856) Anticarcinogenic (0.804)	Cytotoxic activity Agent for Alzheimer’s disease	[303,310]

* Only activities with Pa > 0.5 are shown.

**Table 12 marinedrugs-19-00650-t012:** Biological activities of tellura steroids.

No.	Discovered Activity, (Pa) *	Reported Activity
**129**	Atherosclerosis treatment (0.977) Antioxidant (0.963) Antihypercholesterolemic (0.956) Antiparkinsonian (0.955) Neurodegenerative diseases treatment (0.954) Alzheimer’s disease treatment (0.940) Antihyperlipoproteinemic (0.811)	Activity not studied
**130**	Antihypercholesterolemic (0.958) Atherosclerosis treatment (0.890) Alzheimer’s disease treatment (0.838) Antioxidant (0.807) Neurodegenerative diseases treatment (0.801)	Activity not studied
**131**	Antioxidant (0.922) Atherosclerosis treatment (0.908) Neurodegenerative diseases treatment (0.877) Antihypercholesterolemic (0.869) Alzheimer’s disease treatment (0.868) Antiparkinsonian (0.848)	Activity not studied
**132**	Antihypercholesterolemic (0.909) Atherosclerosis treatment (0.876) Alzheimer’s disease treatment (0.828) Neurodegenerative diseases treatment (0.808) Biliary tract disorders treatment (0.807)	Activity not studied

* Only activities with Pa > 0.5 are shown.

**Table 13 marinedrugs-19-00650-t013:** Biological activities of astatosteroids.

No.	Discovered Activity, (Pa) *	Reported Activity
133	Antihypercholesterolemic (0.967) Antineoplastic (0.824) Bone diseases treatment (0.796) Hypolipemic (0.785) Neuroprotector (0.758) Antipsoriatic (0.739) Anti-inflammatory (0.728) Apoptosis agonist (0.724) Prostate disorders treatment (0.719)	Activity not studied
134	Antihypercholesterolemic (0.927) Bone diseases treatment (0.784) Hypolipemic (0.740)	Activity not studied
135	Antihypercholesterolemic (0.920) Growth stimulant (0.805) Bone diseases treatment (0.743)	Activity not studied
136	Antihypercholesterolemic (0.901) Bone diseases treatment (0.720) Growth stimulant (0.703)	Activity not studied
137	Antihypercholesterolemic (0.912) Bone diseases treatment (0.777)	Activity not studied

* Only activities with Pa > 0.5 are shown.

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
