# Peer review of "In Silico Prediction of Steroids and Triterpenoids as Potential Regulators of Lipid Metabolism"

_marinedrugs, 2021, doi:10.3390/md19110650_

Round 1

Reviewer 1 Report

In the manuscript entitled “In silico Prediction of Steroids and Triterpenoids as Potential Regulators of Lipid Metabolism”, the authors reviewed steroids and triterpenoids with the potential to regulate lipid metabolism. For individual steroid or triterpenoid, 3D drawing of the predicted biological activities was presented. However, there are some issues needed to be clarified or modified.

1.There are some grammatical and writing errors in the manuscript.

2.There are 13 tables and 30 figures in the manuscript, please further summarized and combined.

3.There have been some reviews on steroids with the potential to regulate lipid metabolism. Please show the differences from these reviews, that is, the importance and novelty of this review.

4.The references 21 and 27are the same.

Author Response

Thanks to Reviewer 1 for reading the manuscript and for some guidance.

All found grammar and writing errors have been corrected.

Your recommendations to summarize and combine 13 tables and 30 figures is not possible.

So, these are completely different chemical structures and they cannot be combined.

Several reviews have been published on a similar topic, but this review has summarized data from more than 300 publications and made conclusions. This is the novelty of the presented review.

All changes, additions, and replacements of sentences, letters or numbers are highlighted in green right in the text.

Reviewer 2 Report

This review by Dembitsky reported the prediction of steroids and triterpenoids as potential regulators of lipid metabolism. The chemical structure, source, and reported biological activities of a total of eight groups of steroids and triterpenoids were reviewed. The potential of these steroids and triterpenoids as regulators of lipid metabolism was predicted by using the well-known computer program PASS. This review is of interest to researchers focusing on the regulation of lipid metabolism.

Major comments:

In the Introduction part, the important role for steroids and triterpenoids in lipid metabolism should be reviewed in more detail.

Another major issue of this review is that the presentation of many sentences is quite confusing; these sentences should be rephrased. Only a few examples were listed below.

  1. Lines 200-201, what does the author mean for the sentence “…and their the 3D graph the predicted and calculated anti-hypercholesterolemic activity…”? May the author want to present that “…and their 3D graph showing the predicted and calculated anti-hypercholesterolemic activity…”?
  2. Line 202, for the sake of clarity, the comma in the “in Fig. 4” should be revised to period.
  3. Line 249, “…, such testosterone...”, may the author want to present “ such as testosterom…”?
  4. Line 277, the comma before “in addition” should be revised to period.
  5. Lines 282-284, the sentence is quite confusing, please rephrase it.
  6. Lines 348, add “of” after “activity”.
  7. Line 371, may the author want to say “the α,β-epoxy steroids produced by some freshwater and marine invertebrates…”?
  8. Lines 439-441, rephrase “, and more interestingly, the ethereal extract contains…” to “. More interestingly, the ethereal extract of Arum italicum contains…”
  9. Line 492, what did the author mean about “ an isolated compound”.?
  10. Line 507, revise “was” to “were”.
  11. Rephrase lines 616-617.
  12. Line 651, rephrase “… that do not show natural counterparts…”.
  13. Rephrase lines 664-666 and lines 668-671.

Other points:

  1. Line 287, the number “31”, line 470, the numbers “59, 63, 66” should be in bold format.
  2. Line 777, cite reference for “In the literature, agents that ….are quite fully described.”

Author Response

Thanks to Reviewer 2 for reading the manuscript and for some guidance.

All comments made by the reviewer have been corrected. In the introduction, the role of steroids and triterpenoids in lipid metabolism is presented in more detail. Many sentences have been rephrased or rewritten.

All remarks on points have been corrected. All changes, additions and replacements of sentences, letters or numbers are highlighted in green in the text.

Reviewer 3 Report

The manuscript 1455893 contains some mistakes in the text and in steroid names and drawings. The author must follow the IUPAC Nomenclature Rules for Steroids. An attached file contains some error examples and proposed modifications.

Author Response

Thanks to Reviewer 3 for reading the manuscript and for making many recommendations.

He did a great job. All found grammatical errors have been corrected.

All points in the comments of Reviewer 3 have been corrected. The chemical structure of 46 has been corrected, and 41.

References numbered 11 and 199 were replaced, and numbered 78, 172, and 196 were left unchanged, as they carry certain information. Some phrases and sentences have been changed or rewritten.

 All changes, additions, and replacements of sentences, letters, or numbers, as well as new sentences and paragraphs, are marked in green in the text.

Round 2

Reviewer 1 Report

I have no further questions.